# A Geometrical Structure-Based New Approach for City Logistics System Planning with Cargo Bikes and Its Application for the Shopping Malls of Budapest

**Dávid Lajos Sárdi *** and **Krisztián Bóna**

Department of Material Handling and Logistics Systems, Budapest University of Technology and Economics, 1111 Budapest, Hungary; krisztian.bona@logisztika.bme.hu
* Correspondence: david.sardi@logisztika.bme.hu

**Featured Application: In our paper, we developed a new, geometrical structure-based approach and its graph theory-based geometric model with graph theory-based notation, which can be used in the future for city logistics system planning with cargo bikes.**

**Abstract:** Nowadays, cargo bikes are seeing an ever-greater role in city logistics with an increasing number of deliveries, and it is essential to examine their future role in green and smart cities. In our work, we examine the application of cargo bikes in the city logistics system of the urban concentrated sets of delivery locations, focusing first on shopping malls, with the investigation of the geometrical structure of the logistics network. In the examined concept, the use of cargo bikes will be combined with electric trucks to make possible green deliveries of urban concentrated sets of delivery locations. In this paper, we present the experiences of the existing systems and the related research, the simulation model of the examined new concept with cargo bikes and its results, the graph theory-based geometric model of the examined city logistics system with graph theory-based notation, and the application of the new approach for Budapest. The main output of this research is the geometrical model of the urban concentrated sets of delivery locations and its application. Based on this geometrical model, it will be possible to decide about the suitability of the examined cargo bike-based city logistics concepts for given cities.

**Keywords:** city logistics; cargo bike; cargo cycle; simulation; geometrical model; shopping mall; concentrated sets of delivery locations; bike delivery





## 1. Introduction

In the City Logistics Research Group of the Department of Material Handling and Logistics Systems at the Budapest University of Technology and Economics [1], we have focused on the logistics systems of the so-called urban concentrated sets of delivery locations (abbreviated as CSDL) since 2015. After data collection, development of several models, and complex analysis, we started a project to examine how to use cargo bikes in the new, innovative city logistics systems of shopping malls—one of the most critical CSDLs—and the first studies have shown that it is worth focusing on this area. In the introduction, the definition of the CSDLs, the purpose of the research, and the structure of the paper can be seen.

In the approach of the City Logistics Research Group, the urban delivery locations have two main groups. There are single delivery locations and concentrated sets of delivery locations (urban zones or buildings with a high density of delivery locations), which are groups of single delivery locations with different concentration degrees [2]. Based on our previous research results [3], there are two main types of concentration: concentration with either open or closed infrastructure. In the case of open concentration, the set is marked by an open area, so the stores are surrounded by roads and squares, such as in the case of a

shopping or pedestrian area surrounded by urban roads or an open market surrounded by a square. Within a closed infrastructure, the CSDL is marked by a building that makes a set from the single delivery locations. These main groups of concentrated sets of delivery locations have several types (subgroups), as can be seen in Figure 1. It can be concluded that this classification and the characterization of the properties of the typical delivery locations are also significant results of this research. This classification can be important for city logistics system planning nowadays and in the smart cities of the future as well; as for the deliveries of the CSDLs different solutions are needed than in the case of the single delivery locations.

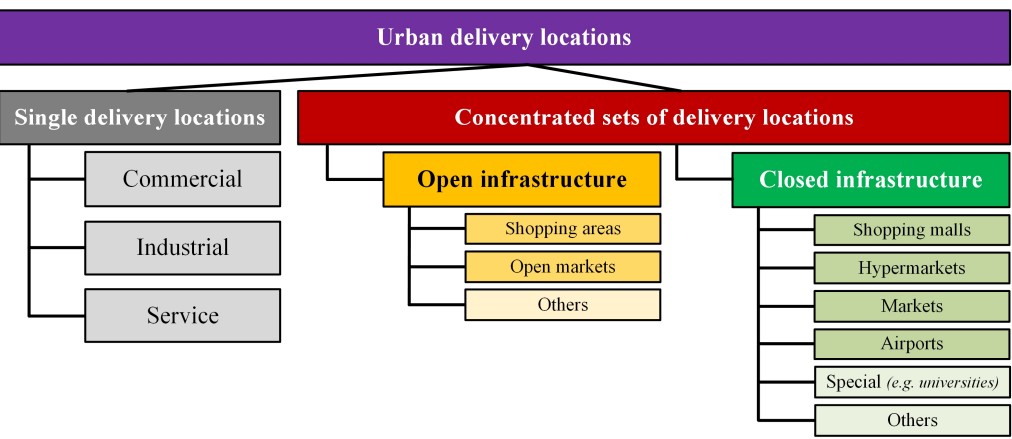

**Figure 1.** Groups of urban delivery locations [3].

As the first step of this research project, a complex data collection methodology was developed [4], which helped to collect data about 5 CSDLs from Budapest (4 shopping malls and 1 shopping area) and 540 stores within them. In Budapest, there are 32 shopping malls with a sum of 3432 stores. In the data collection phase, 4 of these 32 shopping malls were examined, with a sum of 663 stores (19% of all the stores from shopping malls in Budapest), and 377 of them answered our questionnaire (57% of the examined stores and 11% of all the stores of shopping malls in Budapest). Additionally, we examined the most significant shopping area of the city (the "Váci utca" shopping area), with 418 stores. In this research, 163 of them answered our questionnaire (39% of the stores of the shopping area). The collected data provided input for the modeling, where a complex mathematical model [3], a mesoscopic simulation model for the examination of the physical parameters and the cost structure [5], and a topological model for shopping areas [3] were developed. Based on the results provided by the simulation, the consolidation-based, multi-stage city logistics solutions can result in significant savings in the examined systems, both in terms of performance indicators and costs.

The field understudy is significantly important because in every former research, only some sub-systems of the systems were examined, and not with the analysis, simulation, and optimization of the whole systems with several different types of CSDLs. Much research examined the consolidation centers, the cross docks, and the technological solutions, and some projects focused on some well-defined types of CSDLs. In one project, the optimal position of shopping malls and logistics centers were examined based on performance and cost parameters [6], and in another one, the loading and parking areas of a shopping mall were in focus [7]. In some cities, there are existing consolidation-based systems in the city logistics system of (primarily) shopping areas, e.g., in Bristol, Padova, and Nijmegen [8], and in Budapest, there was such a system in the 1970s. The implementation of these solutions led to significant savings, and their results were used as inputs.

In the research of our Research Group, several innovative, gateway-concept-based city logistics solutions were developed. We examined fuel oil-powered small trucks, green trucks, and vans, the use of urban railways and cargo boats; the suitability of the metro

network for freight transport was examined too, and cargo bikes had different roles in several solutions, such as deliveries, home deliveries, or deliveries between stores. In this paper, we would like to examine a concept of serving the shopping malls of Budapest using small green trucks and cargo bikes, based on the investigation of the geometrical structure of the logistics network [9]. The primary purpose of the paper is to present this new concept, which is basically a new city logistics approach, where cargo bikes are combined with electric trucks in a city logistics system of concentrated sets of delivery locations, based on the geometrical structure of the logistics network. In our paper, we show how this new approach can be examined and described. It was expected that such a solution could help to reduce emissions in the examined city logistics system and to reduce the load of the urban roads at the same time. It can be assumed, too, that such a solution can be more expensive than a simple city logistics system with direct, consolidated deliveries, but regarding this question, it is very important that in the case of any traffic issues, the cargo bikes can have better possibilities to perform the delivery tasks. To show these expected effects is an essential purpose of this paper.

For this, first, we present those existing city logistics solutions and previous research projects which provided data, ideas, and experience for our work. From these, those projects were the most important where the combined use of cargo bikes and lorries were examined. Next, a complex data analysis will be presented for the concentrated sets of delivery locations of Budapest to show the usage possibilities of cargo bikes in their city logistics system. In the case of this data analysis, 327 stores from 3 shopping malls (while in the sum 32 shopping malls of the city there are 3432 stores) and 163 stores of the biggest shopping area of Budapest (where the sum number of stores at the data collection was 418) were examined, as in their case, we had available data about the cargo bike delivery-related questions. After the data analysis, we present a macroscopic simulation model to examine this new city logistics approach in the case of all 32 shopping malls of Budapest, and the results of the simulation will show the expected effects of the examined approach with cargo bikes and electric trucks. At the simulation, the most important purpose is to get data about the expected number of delivery vehicles, about the performances, and about the expected delivery costs of the examined system. Before the simulation, it was necessary to estimate the demands of the unknown shopping malls as well; for this, an ABC-analysis-based new solution was used, which will be presented in this section, too. After these, we show the relevant urban structures and the theoretical, symmetrical, and real geometrical model of the examined city logistics system, with a graph theory-based notation and its application for the simulated city logistics system from Budapest. In the last section of our paper, we write about the next steps of the research, where the possible reference structure of the geometrical model must be highlighted, and the use of a multicriterial model will be very important as well to make possible for any examined city a decision about the use of the new approach.

The main sections of the paper are the followings:

- Materials and Methods: use of cargo bikes in city logistics systems—presentation of existing cargo bike projects, previous research experiences, data analysis for Budapest regarding the possible city logistics use of cargo bikes, the simulation model of the examined city logistics system, and results of the simulation.
- Results: the theoretical geometrical model and its application for Budapest—presentation of the related urban structures, the theoretical, the symmetrical, and the real geometrical model with graph theory-based notation and its application for Budapest.
- Discussion and future research—discussion of the results and presentation of the next steps of the research, highlighting the analysis of the reference geometrical structure and the development of a multicriterial ranking model for decision support.
- Conclusions—summary of the paper and its conclusions.

## 2. Materials and Methods: Use of Cargo Bikes in City Logistics Systems

Cargo bikes in city logistics systems are becoming more and more popular, and they can be useful in smart cities as well, as they are very flexible, and it is very simple to integrate them into any city logistics solution. In addition to single two-wheeled cargo bikes, bigger equipment and special trailers can be used, which are able to deliver even pallet-sized unit loads. Most of them have electric power assistance, so they can be used for bigger distances and for larger amounts. Their examination is critical, similarly to every other electric and $CO_2$-free solution (like electric lorries, drones, or electric cargo mopeds), as getting a $CO_2$-free urban freight transport is one of the main purposes of the European Union in this field [10]. Electric vehicles are in the focus of current research projects as well, but these projects are examining mostly the usual road delivery solutions [11]; their costs, their range, charging times, and their capacity is essential as well [12], and it became necessary to develop new city logistics concepts in this field. It is also clear that the number of electric vehicles in city logistics systems is increasing, and they have similar benefits as the cargo bikes, but as they have a long charging time and their range is limited, a significant drawback can be concluded in comparison with the most used urban vehicles [13]. In contrast, in the case of cargo bikes, the batteries are combined with human power, and congestion is not a problem for them, so it is important to examine them in these new city logistics concepts. Next to cargo bikes, delivery drones are examined in several research projects as they are getting more and more popular, and they can be used for more and more tasks in city logistics [14,15], but if we compare cargo bikes to them, there will be still several advantages: batteries are still combined with human power, landing is not a problem for cargo bikes, they have bigger capacity, and the flight regulations do not mean restrictions for cargo bikes. Based on these, it can be concluded that cargo bikes have several advantages if they are compared with the other electric urban delivery solutions, so it is very important to calculate with them in the case of any city logistics developments.

### 2.1. Existing City Logistics Systems with Cargo Bikes

Based on studies, nowadays, in most European cities, every second motorized travel in freight transport could be handled by bikes instead of passenger cars and lorries [16]. In the case of home deliveries, every fourth package could be delivered by two-wheeled utility bikes [16]. Accordingly, such systems are operating in a large number of European cities. For example, DHL works in several cities with cargo bikes. We would like to highlight their cubicycle system, a four-wheeled construction [17], which is used in several cities in the Netherlands (e.g., in Utrecht), and additionally in Gent, Frankfurt, and Taipei. In these systems, EUR-pallet-sized (EUR-pallet: 800 mm × 1200 mm) intermodal units are delivered by lorries to transshipment points where they are passed over by a special equipment to the cargo bikes (Armadillo cubicycles), and they solve the LTL (less thank truck load) last-mile deliveries of the delivery locations of their zone. In the geometrical model-based new approach, which is presented in this paper, the deliveries are organized in a similar way, with more stages and transshipment points, but the routes will be selected based on the geometrical structure of the network.

Another city logistics system that should be highlighted is Eadessopedela from Rome, Italy. In this system, from a so-called "mobile storage" the goods are loaded to cargo bikes at a higher point of the city, and they go to the delivery locations by using the advantages of the terrain [18]. This mobile storage-based concept is dynamic, so the performances and the emissions can be minimized. A similar system is operated by UPS in Hamburg, Dublin, and Leuven [19], and TNT had a similar pilot project in Brussels [20]. In both cases, an intermodal unit functioned as mobile storage, and the cargo bikes were loaded from them.

Additionally, several city logistics solutions can be mentioned: At Vanapedal, in Barcelona, two- and three-wheeled vehicles are used; at Zolle in Rome, only fruits and vegetables are delivered; and the logistics system of Outspoken in Cambridge handles last-mile deliveries and postal mails. In the KoMoDo pilot project in Berlin [21], Hermes operated with cubicycles with special trailers, so it became possible to handle two EUR-pallet-sized intermodal units in their last-mile deliveries [22].

It may also be interesting to examine projects where cargo bikes are combined with other green transportation modes. In Paris, in the Vert Chez Vous system [23], barges deliver cargo trikes on the Seine, and the trikes are loaded with small packets on the boat during the delivery. As a result of this solution, 15 trucks can be replaced each day [24]. DHL has a combined solution in Amsterdam with small cargo boats and cargo bikes, and in Frankfurt, in the LogistikTram pilot project, cargo bikes were combined with cargo trams [25]. In this solution, special logistics units were loaded to the bikes from the trams at transshipment points.

Based on the analysis of 50 city logistics projects, a study concluded that cargo bike-based city logistics solutions could lead to high profit. The authors of this paper highlighted that their result is coherent, and other studies proved the same advantages, not only from an economic point of view but also in terms of the environment and the quality of life [26].

As in our former project, from where the graph theory-based idea comes, primarily Budapest was examined, so the existing cargo bike solutions of the city are important as well. In the city, the length of the bike path-network was 187 km before the pandemic in 2020 [27], a bike-sharing system is available, and there are approximately ten logistics provider companies that use utility bikes for their deliveries. Only some of the active providers are operating cargo bikes or cargo trikes, and significant development potentials can be expected in this field. The biggest logistics provider with cargo bikes in the city is Hajtás-Pajtás [28], which is operating utility bikes, two- and three-wheeled cargo bikes, and an electric lorry, and is organizing deliveries to the city center from a micro consolidation center.

### 2.2. Summary of Research on Cargo Bikes

Naturally, as cargo bikes are becoming more widespread, several research projects have examined cargo bike-based city logistics systems. Most of these studies deal with their application possibilities and with the possible environmental and financial effects of the application. Based on a study from 2014, most of the available publications and documents are examining cargo bikes in Europe, and they are looking for the potential application fields of them; but it is generally argued that cargo bike solutions are a viable alternative for the future [29]. It is also interesting that cargo bikes can be considered as an alternative to walking in the case of the last section of urban deliveries between the loading area and the customer, similarly to drones [30]. A study from 2014 concluded that if the costs, payloads, and range are examined, (electric) cargo bikes are between simple bikes and passenger cars [31], which also have an important role in urban transportation and in urban logistics.

Most of the studies concluded that motorized delivery transactions should be replaced by new solutions, which is why cargo bikes are getting more attention. Based on a paper from 2012, it was primarily the smaller companies that were working with cargo bikes in the CEP-sector (Courier, Express, and Parcel), and their application was motivated mostly by the problems of urban transport and by green aspects [32]. Nowadays, most of the bigger city logistics providers (like DHL or GLS) are expanding their roles in cargo bike deliveries.

Based on a paper from 2017, approximately 10% of lorries and trucks could be replaced nowadays by cargo bikes (without decreasing efficiency) in urban areas where there is a maximum of 2 km linear distances. This could reduce $CO_2$ emissions significantly, and the external costs could be reduced by 25%. Based on these, cargo bikes could be real alternatives for the transport situation of overcrowded and severely polluted city centers [33]. In another paper from 2019, cargo bikes with electric power assistance can be more cost-effective than lorries, in the case of small amounts, in densely populated urban areas [34]. In this paper, parking issues were highlighted as well, with cargo bikes having a huge advantage in this field over lorries.

As modeling is the primary focus of this paper, the most significant studies are those where the development and application of models for cargo bike-based city logistics systems are in focus. These models can have several different purposes, and they use several different objective functions. Some models can compare given scenarios, while in some others, they are trying to find the best solution to a problem by optimization. A study shows that the growth of the shares of CEP deliveries and home deliveries gives an extra reason to examine this field. In this paper, the situation of Munich was discussed, the optimal position of the centers was sought out with a view to optimal cargo bike routes, and the necessary delivery times and the effects of the new system were examined. In this paper, not only the mathematically optimal position for the depo was highlighted, but the greenest and most sustainable position, too. It was concluded that the sum distance could be reduced by integrating cargo bikes into the system, while the delivery times could still be met [35].

In a paper from 2011, the application of cargo bikes and micro consolidation centers in Manhattan (New York) was examined. In the investigated area, there are more than 100,000 delivery transactions each day, and based on their results, a significant part of them could be handled by electric cargo trikes—their integration for last mile deliveries would not cause an increase in costs, while the external effects could be reduced [36]. In a paper from 2017, the application of consolidation centers and cargo bikes in Sao Paulo was examined. In this city, TNT is currently operating cargo trikes for city logistics, while on most of the routes, there are problems with the use of lorries because of congestion, so the advantage of cargo bikes could be taken here, too [37].

Examining the case of London, the benefits of the application of cargo bikes are the reduction of vehicle purchase, delivery and parking costs, faster deliveries in congested city centers, and the reduction of emission. As a disadvantage, safety aspects and limited capacity were highlighted [32]. The expected benefits can be confirmed by the results of the pilot project from London in 2010. In this project, 7 fuel oil-powered lorries were replaced by 6 cargo trikes, 3 electric lorries, and 1 fuel oil-powered truck, adding a consolidation center to the system. In this project, in the urban areas, the mileage was reduced by 20%, and $CO_2$ emission was decreased by 54% [38].

In a study from 2019, the case of Seoul was examined, and similarly to our project, a city logistics system with cargo bikes and lorries was modeled. In this model, the share of cargo bikes and lorries was handled as an important strategic decision, and they analyzed scenarios by use of VRP algorithms (Vehicle Routing Problem) for the whole city. In their model, simulated annealing was used, and in the scenarios, more and more lorries were replaced by cargo bikes. Based on their results, the sum costs can be significantly reduced by adding cargo bikes; but it is also a significant result that not every lorry should be replaced by bikes because of their capacity. They concluded that $CO_2$ emission could be reduced approximately by 10% [39]. In a study from 2016, the combined application of cargo bikes and lorries was examined in Vienna. In this project, a two-stage model was developed, in which the different delivery vehicles are synchronized. They concluded, too, that emissions can be reduced by integrating cargo bikes. However, they say that the delivery costs can be increased [40]. A paper from 2018 focused mainly on restaurant deliveries by cargo bikes, examining them by simulation. In this research, lorries were added to the examined logistics system, which delivered the goods to urban consolidation

points, and the last-mile delivery tasks were handled by cargo bikes. In this model, there is a new objective next to the minimization of the sum mileage: the minimization of the sum delays [41], which is very important in the case of delivering foods. The authors of this study published a simulation-based decision support system, too, and their results draw attention to the fact that, in these kinds of systems, it is essential to appropriate the number of available cargo bikes [42]. In another paper from 2017, a discrete-event-simulation-based (DES-based) solution was developed for the examination of the use of cargo bikes in multi-layer logistics networks, and the model was adapted for Grenoble. This model provides results about the daily time needs, demands, and the number of deliveries. Based on the results, it was concluded that $CO_2$ emissions could be reduced in this concept as well, and urban traffic jams will also be reduced [43].

The results of former research projects show that the city logistics use of cargo bikes can be various, and, from the modeling point of view, this area can be approached in many ways. Comparing the results of several projects, it can be concluded that in the case of the integration of cargo bikes, emissions and congestions could be reduced. However, mixed results can be seen for the different cost components: some of them can be reduced, but others are increasing in the new concepts. Based on the results of the literature review, it can be concluded that the geometrical model and the graph theory-based approach will be a completely new research direction, but several items from the results of previous research can be used, especially from models which examine lorries and cargo bikes together. It is also worthwhile to highlight that a graph-based approach to the cargo bikes-based city logistics systems would also be new, even though such an approach is used in transport and logistics research, for example, in the case of transport networks [44,45], inverse logistics aspects [46] or maintenance aspects [47].

Next, we would like to present the cargo bike-related results of our data collection, and we would like to summarize the results of the simulation project which motivated this paper.

### 2.3. Possible Use of Cargo Bikes in the City Logistics System of Concentrated Sets of Delivery Locations in Budapest

In the first phase of the research, 5 CSDLs in Budapest were examined, and we received data from about 540 stores by use of our own questionnaire [4], so the main logistics properties of 4 shopping malls and the properties of the stores of the biggest shopping area of the city are known. As in the case of the fourth shopping mall, not every cargo bike delivery-related question was included; only the data of 490 stores (327 stores from 3 shopping malls and 163 stores from a shopping area) are examined in this section.

First, we would like to highlight data about the delivery units. This question was answered by 475 stores of the 490. Of these, 417 (87.8% of the responders) are using (mostly small) boxes, and their quantity, size, weight, and volume are various—these are the most non-standard units in these systems. In one delivery transaction, an average of 16.6 boxes are arriving (standard deviation, 25.8), 5% of them are under 1 kg, 49.8% are between 1 and 5 kg, and 45.2% are more than 5 kg. This means a sum of 6 kg of goods in boxes every day per store (st. dev., 19.7 kg; yearly sum, 2.19 tons of goods in boxes/store). Most of these could be handled by cargo bikes, as the amount and their weight are mostly small. Of the responding stores, 12.8% are using pallet unit loads (there are stores that are using more delivery unit types); in one delivery, an average of 4.6 pallet unit loads are arriving (st. dev., 5.6). Of these, 42.9% are under 100 kg, and another 50% are under 500 kg, so most of these units could be handled by the three- or more-wheeled cargo bikes with a bigger capacity. Additionally, other, mostly small delivery units were mentioned by the stores, as can be seen in Figure 2.

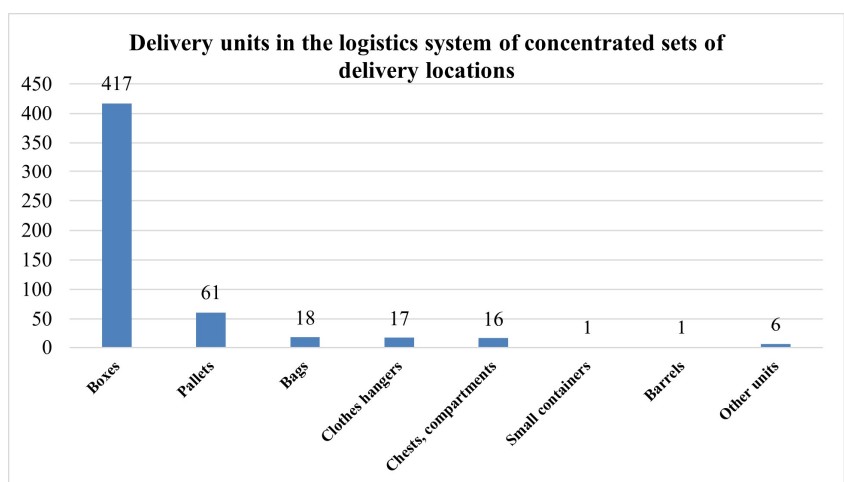

**Figure 2.** Delivery units in the logistics system of concentrated sets of delivery locations (*N* = 475).

The next important question was about the delivery vehicles currently being used. Most of the 474 responder stores are using lorries (277 stores, 58.4%) and their own passenger cars (171 stores, 36.1%) for the deliveries. These smaller vehicles could be easily replaced by the different types of cargo bikes in most cases, as most of the delivery units are small. Despite all of this, currently, only 5.1% of the stores (24 stores) are using bike or motorbike couriers for their deliveries.

It was examined what share of stores assumes that a given cargo bike (BULLITT two-wheeled bike, with a 50 × 50 × 70 cm delivery box) would be appropriate for their logistics transactions. Of the 375 responder stores, 29.3% answered that based on the size of their logistics units, the cargo bike could be used. Of these, 50% (55 stores) answered that the cargo bikes could be used for their incoming deliveries, and in the case of 20% (22 stores), they responded that the bikes could be used for home deliveries. A figure of 41.8% (46 stores) suggested that cargo bikes could serve the deliveries between their stores (see Figure 3). In the case of the incoming deliveries, an average of 3.91 daily deliveries by this cargo bike type would be necessary (st. dev., 5.29), and a maximum of 25 deliveries would be enough. In the case of 76.1%, a daily maximum of 3, and if 91.3%, a maximum of 10 daily deliveries would be enough. Based on their answers, 180 deliveries each day would be necessary to handle the demands of the 46 responding stores.

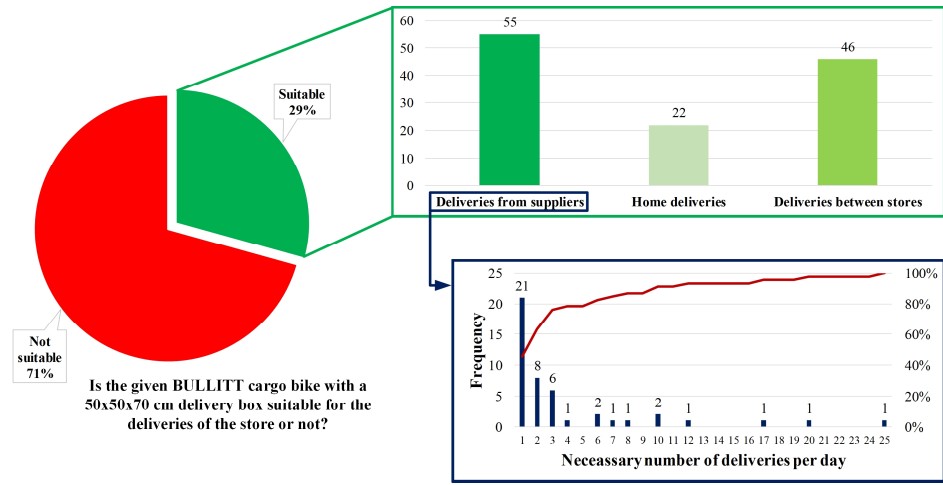

**Figure 3.** Suitability of a BULLITT cargo bike for different delivery tasks (*N* = 375).

It is essential to highlight here that this question was about one of the smallest available cargo bikes, so if the bikes with a bigger capacity were examined, the results would be even

better, and a bigger suitability share could be seen. Based on these results, though, in the examined city logistics system, cargo bikes can be used for several different purposes. In this paper, we are going to discuss their integration into the delivery process of the CSDLs, in a well-defined, geometrical model-based city logistics concept. Next, we would like to present the simulation model from our former project and its results [9], which gave us the idea of this paper.

### 2.4. Application of Cargo Bikes in the City Logistics System of Shopping Malls in a Geometrical Model-Based Approach

Based on the previous section, it became clear that cargo bikes can become a part of the city logistics system of the CSDLs; only the deliveries are examined here, as they are the most significant logistics tasks. In a project in 2017, 18 shopping malls from Budapest were examined, together with a city logistics provider. In this project, it was expected that the existing logistics areas of the examined malls are appropriate to handle all the logistics tasks in the examined concepts, and it was also assumed that the shopping malls and the stores could work together in the new system. In this project, a geometrical model-based approach for city logistics modeling was developed. For this, two main groups were defined for the shopping malls (for the CSDLs, as the same concept can be suitable for any type, e.g., markets or shopping areas): there are malls on the central ring and malls on the rays that start from the central ring. In the simulation runs, it was examined how it is possible to integrate the deliveries of some stores of the shopping malls on the rays by use of cargo bikes (or use of electric lorries) to the deliveries of the malls on the central ring, in a gateway-concept based city logistics system. The primary purpose was to calculate the necessary number of vehicles in the examined concepts. The essence of the examined geometrical model-based solution is that instead of using VRP in the model, the geometrical model defines the locations, the connections, and the routes based on predefined rules, so the logistics network is given.

In this project, 4 different concepts were examined. In these concepts, the deliveries have been realized from the consolidation center next to the external ring of the city defined by the geometrical model, by the use of bigger electric trucks (EMOSS CM12 e-trucks in the simulation) to the central ring, so the central shopping malls have been served in this way. To these deliveries, the goods of those stores of the shopping malls will be added on the rays, in which case, the green small delivery vehicle on the rays (defined for each concept) is appropriate, based on the amount and size of goods. In the concept, these goods will be delivered from the central malls to the malls on the rays, using cargo bikes or electric lorries. The concept can be seen in Figure 4.

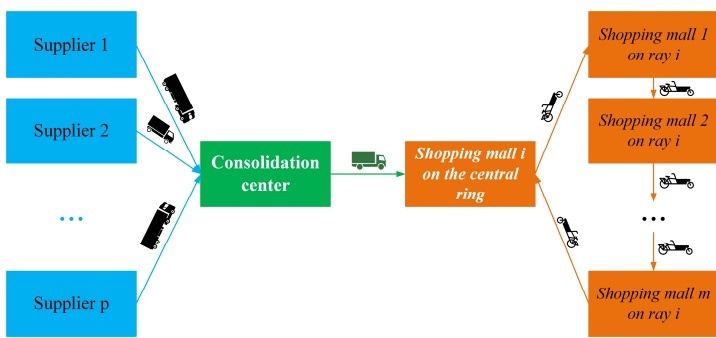

**Figure 4.** Examined new city logistics concept with cargo bikes.

In the first three concepts, we proposed the use of the previously examined BULLITT cargo bikes (with 50 × 50 × 70 cm delivery boxes) with electric power assistance, choosing those stores on the rays where a daily maximum of 1, 2, or 3 cargo bike deliveries are enough. In the fourth concept, the cargo bikes have been replaced by electric lorries (Nissan eNV-200 in the simulation model), choosing those stores on the rays where 1 delivery per

day by lorry is enough. We developed a macroscopic simulation model to examine these concepts in MS Excel, which worked with aggregated data per shopping mall. The input data were provided by previous data collection; the data for every mall were estimated based on the data of 3 malls and their stores.

In this project, the primary purpose of the simulation runs was to receive data about the necessary number of vehicles in the examined concepts to handle the given delivery tasks. In every case, 12 e-trucks will be enough to handle all the delivery needs, so by adding more and more goods to the system, no more trucks are needed. Thus, with a relatively small investment cost, more and more stores could be served with the use of green vehicles: Additionally, 5, 9 or 13 BULLITT cargo bikes, or 8 electric lorries would be enough in the four concepts, and if cargo bikes are used, the load on the congested roads will become smaller [9].

As these simulation runs were performed for a given project, with given vehicle types, and with given shopping malls to serve, we decided to complete our model with all other malls from Budapest. It was also decided to use a new estimation methodology to provide input data for the model, and it was planned to examine several different types of cargo bikes in the new scenarios (with different sizes and capacities). Previously, only 18 shopping malls were examined and simulated [9], but in Budapest, there are 32 of this kind of CLSD, and most of them fit the previously examined rays or their extensions. All these malls and their geometrical model can be seen in Figure 5. The central ring is black on the figure, the rays start from it, grey means the city border, and blue is the external ring road of the city. Previously, data were collected about 4 of these 32 shopping malls, and in the case of 3 of them, data were collected about the cargo bike delivery-related questions as well.

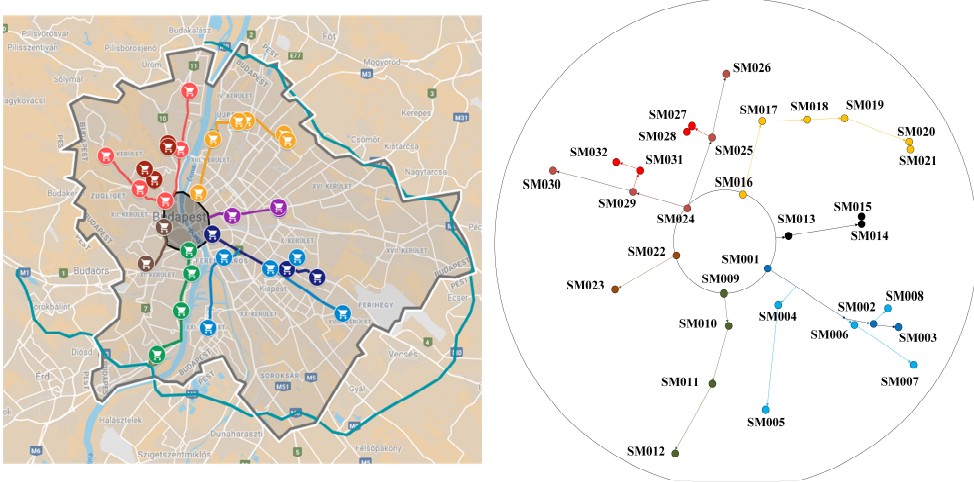

**Figure 5.** The examined shopping malls on Google Maps and the simplified version of the geometrical model.

For the estimation of the demands, an ABC-analysis-based [48] solution was used, where the malls were assigned based on the number of their stores to the categories "small", "medium", or "large". The number of deliveries and the demands was not considered for this solution (as there were no available data for every shopping mall), only the number of stores per shopping mall was examined, but it was assumed that the number of deliveries is proportional to the number of stores, as the types of the stores are similar in case of every shopping mall. The chosen limits were 33.33%, 66.67%, and 100%, so the stores of category A ("large") can give a maximum of one-third of all the stores of the malls from Budapest (and it was assumed that with this they give a maximum one-third of all the delivery demands of shopping malls from Budapest), and category A and B ("medium") can give a maximum two-third of the stores. The results of the analysis can be seen in Figure 6, where

green is category A, yellow is category B, and red is category C. For anonymity reasons, the examined shopping malls are marked by their ID (identification numbers) in the paper.

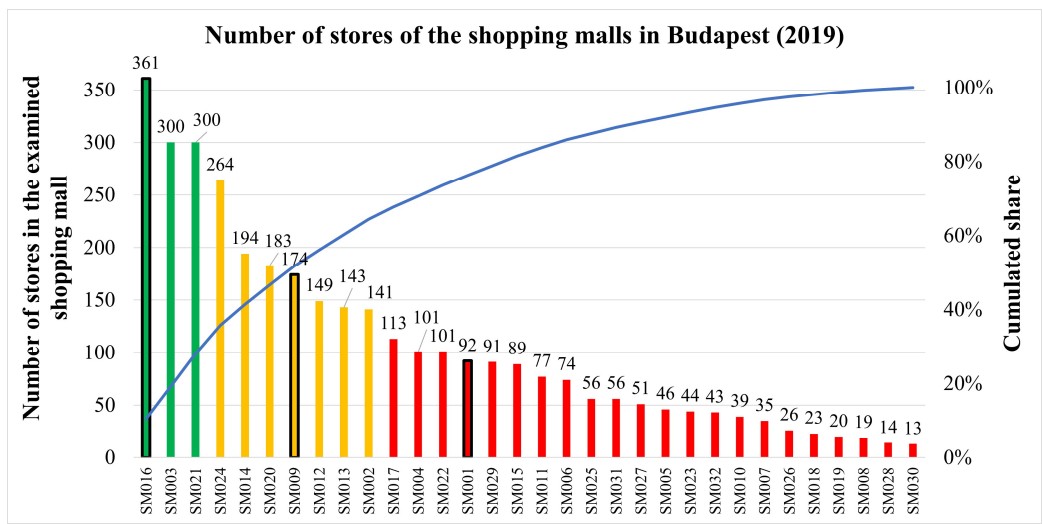

**Figure 6.** The number of stores and the ABC category of the examined shopping malls.

On the chart, the black outline shows those 3 shopping malls for which there were available data. As one of them is falling into every category, they can provide a reasonable basis for estimating the demand in the next steps. Based on the results of the former data collection, the expected characteristics can be seen below for each category:

- Category A: shopping malls with a big number of stores, mostly smaller stores with smaller demands. Their demands are expected to be served easily by cargo bikes.
- Category B: shopping malls with a middle number of stores, mostly stores with a bigger floor area and bigger demands. Their demands are expected more challenging to be handled by cargo bikes, but still, a significant share of the stores can be served by them.
- Category C: shopping malls with a low number of stores, most stores with a bigger floor area and bigger demands. Their demands are expected to be more challenging to be handled by cargo bikes, but there is still a share of the stores which could be served by them.

Based on this analysis, it was concluded that there is no correlation between the categories and the location of the malls in the geometrical model.

Next, the demands of all the malls were estimated based on the real data from its category, proportionally to the number of stores. First, specific values were calculated for all 3 categories based on the real data of the 3 shopping malls. The daily, weekly, monthly, and yearly expected number of deliveries were calculated, as well as the number of goods to be handled per day and per month, in kg and in m$^3$. We also examined with 4 cargo bike types the expected daily and the monthly number of deliveries, the number of stores to be served by them, and the expected daily and the monthly amount of goods to be delivered by them, in kg and in m$^3$. These 4 bike types will provide the 4 scenarios in the extended simulation model. To represent the main size-categories of the cargo bikes, 1–1 two-, three-, four- and six-wheeled solutions were chosen, and their parameters can be seen in Table 1. In the simulation, it was expected that the structure of each bike type is appropriate for the delivered logistics units, and the loading technology is available at every junction of the new system.

**Table 1.** The parameters of the examined cargo bike types.

| Cargo Bike Type | STePS eBULLITT | Radkutsche Musketier Trike | Armadillo Cubicycle | Armadillo Cubicycle with Trailer |
|---|---|---|---|---|
| Wheels | 2 | 3 | 4 | 6 |
| Load capacity [kg] | 100 | 242 | 125 | 250 |
| Load capacity [m$^3$] | 0.245 | 1.339 | 0.960 | 1.920 |
| Number of delivery units | 1 | 1 | 1 | 2 |
| Length of the delivery unit—L [m] | 0.70 | 1.27 | 1.20 | 1.20 |
| Width of the delivery unit—B [m] | 0.50 | 0.83 | 0.80 | 0.80 |
| Hight of the delivery unit—H [m] | 0.70 | 1.27 | 1.00 | 1.00 |
| Battery [kWh] | 0.504 | 0.540 | 1.200 | 1.200 |
| Electric power assistance [W] | 250 | 250 | 250 | 250 |
| Average distance with one charge [km] | 90 | 200 | 50 | 50 |
| Charging time [h] | 4 | 4 | 4 | 4 |

The estimated specific amounts for the 3 categories can be seen in Appendix A (Table A1), and the estimated amounts for the shopping malls (Tables A2 and A3) and the most important parameters (Table A4) can be seen in the tables of Appendix B. These values are macroscopic estimations, and the primary purpose of the estimation was to provide some data to map the real logistics processes, to examine the possible benefits of the examined concepts.

After estimating the demand values, simulation runs were performed by examining all the shopping malls from Budapest, with the geometrical model-based approach (see the geometrical model earlier in Figure 5; this geometrical structure is the one to be examined later). In the simulation model, the consolidation center was placed next to the external ring because of its good transport connections.

In this model, a total of 1135 stores of the 6 central shopping malls are served in each concept. Additionally, there are 26 shopping malls and 2297 stores within them on the rays. In the model, only those stores on the rays which could be served by the given cargo bike type were integrated into the new system (the other stores were not examined in these concepts; we examined only the central malls and those stores which can be added using cargo bikes to the new system). By use of eBULLITT cargo bikes, 342 stores can be served (14.9% of all); by use of Radkutsche trikes, 990 stores (43.1%); in the case of Armadillo cubicycles, this number is 805 (35%); and if a trailer is added to them to get a six-wheeled construction, 1078 stores can be served (46.9%, so nearly half of all stores on the rays). This already shows that by integrating cargo bikes into the system, more and more stores can be served without adding a significant number of extra trucks.

The simulation model was developed as an MS Excel-based macroscopic model, and the primary purpose was to get information about the necessary number of vehicles. A one-month-long period was modeled, and after the experiment design, we got the results that can be seen in Figure 7 for the 4 examined cargo bike types. In the simulation, the e-truck was again EMOSS CM12 type.

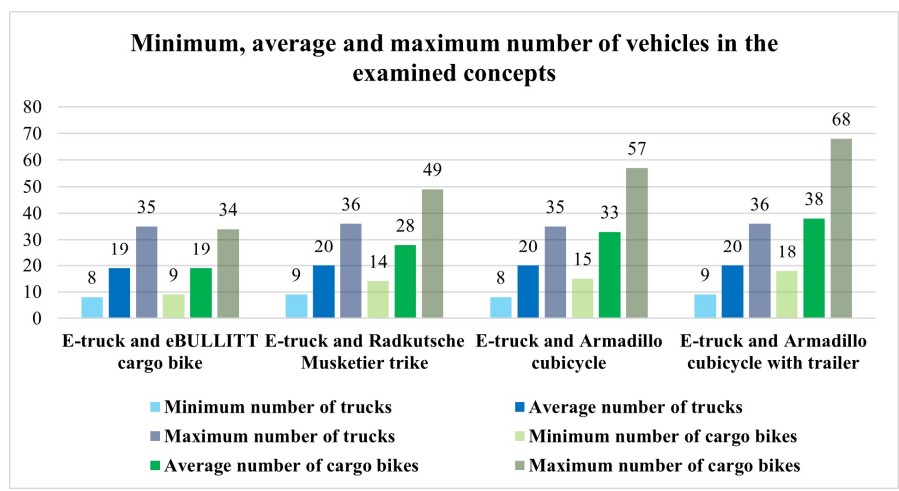

**Figure 7.** The number of vehicles, based on the simulation runs, for Budapest.

The necessary number of vehicles is not critically increasing; adding new stores in the distribution system has impact only on the number of cargo bikes significantly, similarly to the former results from 2017 [9]. If we would like to have a consolidation-based concept, where all the stores of each shopping mall are served by lorries (3432 stores), 60 trucks would be necessary (minimum 36, maximum 86) based on the simulation runs, but in the previously presented concepts, 19–20 trucks (minimum 8–9, maximum 35–36) would be enough, with additional cargo bikes. The necessary number of vehicles has a fluctuation in every case, as the demands are fluctuating in the model (as in the real system), but with organizational solutions, the necessary number of vehicles could be balanced around the expected value.

In the simulation, some other parameters were examined as well; they can be seen in Figure 8.

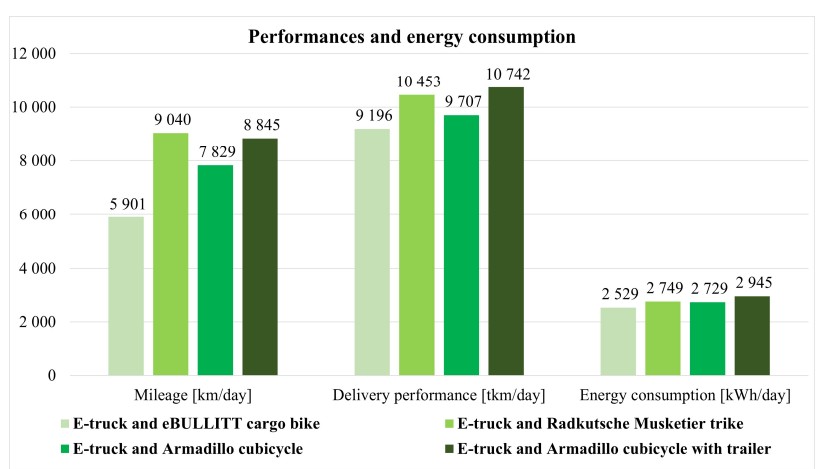

**Figure 8.** Mileage, delivery performance, and energy consumption based on the simulation runs.

Thus, we got positive results for the new system concepts. Based on the demands, a relatively large number of stores on the rays can be integrated into the system using cargo bikes, and the total number of vehicles will not be too large. It is also important that if cargo bikes are used instead of trucks and lorries, it also reduces the load of the overcrowded urban roads, as bike paths can be used instead of them, and the investment and operation costs of the bikes are also smaller. As there is one more stage in this solution than in the case of direct deliveries from consolidation centers to the stores, the delivery costs should be examined as well. It can be assumed that a geometrical model-based solution will not be cheaper, but there are several other factors that should be discussed. Considering the

mathematical model of the cost structure of the concentrated sets of delivery locations [5], the delivery-related costs can be modeled. Based on the simulation runs, these costs were compared for each scenario with the costs of the direct deliveries from the consolidation centers to the concentrated sets of delivery locations by the previously mentioned electric trucks, in the case of the same amount of goods. In this macroscopic level model, the salary of the truck drivers, the salary of the bike couriers, the costs of the energy consumption based on the European Union average price from 2018 [49], and the other maintenance costs of the vehicles was examined. The results can be seen in Figure 9. In the figure, the column-pairs of the direct deliveries and the deliveries with cargo bikes are handling the same amount of goods; the bigger cargo the bike that is used in the system, the greater the amount that can be taken in the direct deliveries. In this comparison, it was assumed that the inventory-related, loading- and material handling-related and administration related costs would not be significantly different if the same amount of goods is handled at the same junctions (and the extra loading tasks in the central concentrated sets of delivery locations can be performed by the bike couriers).

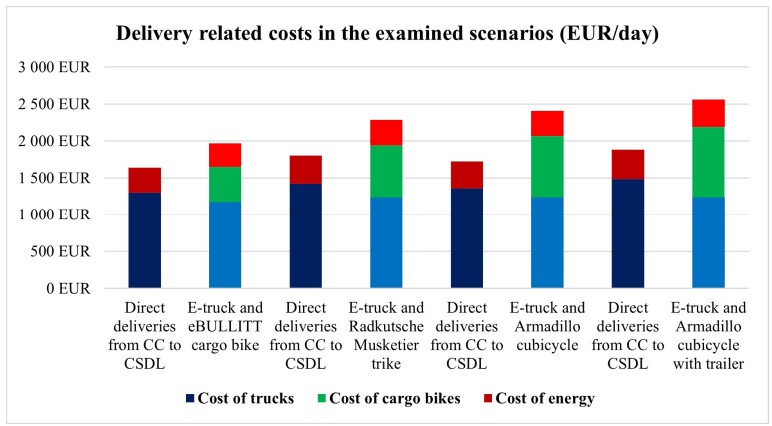

**Figure 9.** Comparison of the delivery-related costs between the examined scenarios and the direct consolidated deliveries.

The delivery-related costs will increase in the case of the solution with cargo bikes, as instead of direct deliveries, there will be an extra junction in the case of the CSDLs on the rays. The increase of the delivery-related costs is between 20% and 39.6% in the reviewed scenarios if the regular operation is examined, and it is more significant if more cargo bikes are needed. If only the financial reasons are considered, it is better to use the direct delivery solution, but if the potential effects of the traffic jams (late deliveries, extra deliveries needed with additional vehicles, etc.) are considered, the currently not significant difference can be even smaller. It must be highlighted as well that nowadays, it is getting more and more difficult to find suitably qualified drivers for lorries and trucks, and it is simpler to employ bike couriers. Especially if the development of the electric power assistance solutions will lead to faster deliveries, the number of required cargo bikes can be reduced, which can reduce the sum delivery costs as well.

So far, we have made estimations using a macroscopic simulation model to examine the effects of the geometrical model-based approach (and the new system concepts based on that). The model showed us how effective it could be to integrate cargo bikes into the examined city logistics system. The simulation provided lots of interesting data regarding this, and based on these results, it can be concluded that in such a city logistics system concept, a significant amount of goods could be handled, and a significant number of stores could be served by integrating cargo bikes, and the load of the urban roads could be reduced. It seems to be clear that the new solution is more expensive than the direct deliveries from the consolidation center to the concentrated sets of delivery locations, but the difference is less than 40% in every scenario, and it is also important that in the case of any traffic issues the cargo bikes can have better possibilities, and by use of the bike

lanes, the urban roads will be less overcrowded. However, there is a significant number of additional questions related to these solutions: how to select the optimal central ring and the rays of the geometrical model, how to assign the CSDLs of the rays to the central ones, and when it is worth thinking about such a system at all. For the last question, we consider it essential to find a reference structure (optimal or close to it) that could be used as a benchmark in the future.

Next, we would like to examine the previously examined and described geometrical model based on graph theory, introducing its new notation. This will be the first step to find the reference geometrical structure, which can make it possible to evaluate cities or urban areas in the future and determine whether it might be worthwhile to examine such a concept in more detail for them or not.

## 3. Results: The Theoretical Geometrical Model and Its Application for Budapest

To describe the geometrical model based on graph theory, first, the relevant urban structures must be reviewed; next, we are going to define the theoretical, the symmetrical, and the real geometrical structure based on graph theory, and in the end, we are going apply the model and the notation to the shopping malls of Budapest as an example.

### 3.1. Urban Structures

To model the city logistics network, it is necessary to know the urban road network and the structure of the examined city. Examined from a morphological point of view, cities are usually composed of several different structures. For the current research, three main structural groups can be identified:

- irregular structure,
- grid-based structure, and
- radial structure.

In the case of an irregular urban structure, irregularity can mostly be found in the whole urban structure; the road network and the arrangement of the building are irregular as well. If the given city or given urban area has resulted from a conscious engineering design, there is a grid-based structure. In the case of radial urban structure, radially converging main roads and rings give the backbone of the urban structure [50], such as in the case of Budapest with "Kiskörút" and "Nagykörút" or in Vienna with the "Ring". The available road network provides a reasonable basis for the organization of the city logistics systems; the road network itself can also establish the logistics network, as the example of Budapest presented it previously. In the current case, the radial urban structure provides the basis of the geometrical modeling since the model that was examined earlier came from the radial structure of Budapest. In the previously examined model, we started from the benefits of the radial structure, taking advantage of the fact that the intersections of the rings and rays designate high-traffic nodes, and the rings and rays themselves define the main urban road network.

In general, there is a tendency for the CSDLs to be located in the main junctions (as they are transport junctions as well, this is ideal for customer traffic), and in the case of a radial structure, these junctions are provided by the intersections of rings and rays. The theoretical geometrical model, which is presented in the next sections, is based on the radial structure, but naturally, it can be adapted to any other structure type if a central ring and rays can also be designated on the road network, as can be seen in Figure 10. Later in the research, it will be an important task to develop the automatic, algorithm-based designation of the central ring and the rays (for example, by use of a genetic algorithm); in this paper, intuitive identification was used.

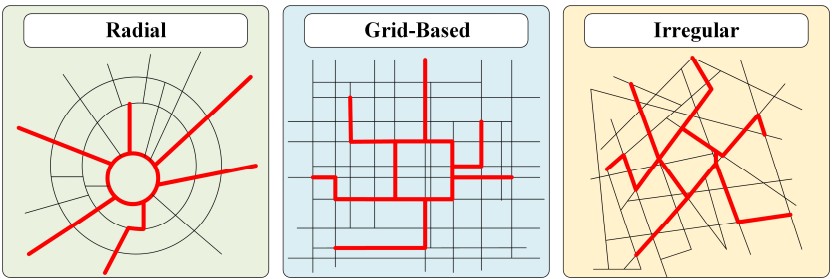

**Figure 10.** Designation of the radial structure-based geometric model in the case of different urban structures.

### 3.2. The Theoretical Geometrical Model

First, the theoretical geometrical model of the city logistics system of concentrated sets of delivery locations (in this case, primarily shopping malls) is defined. This theoretical model can be seen in Figure 11.

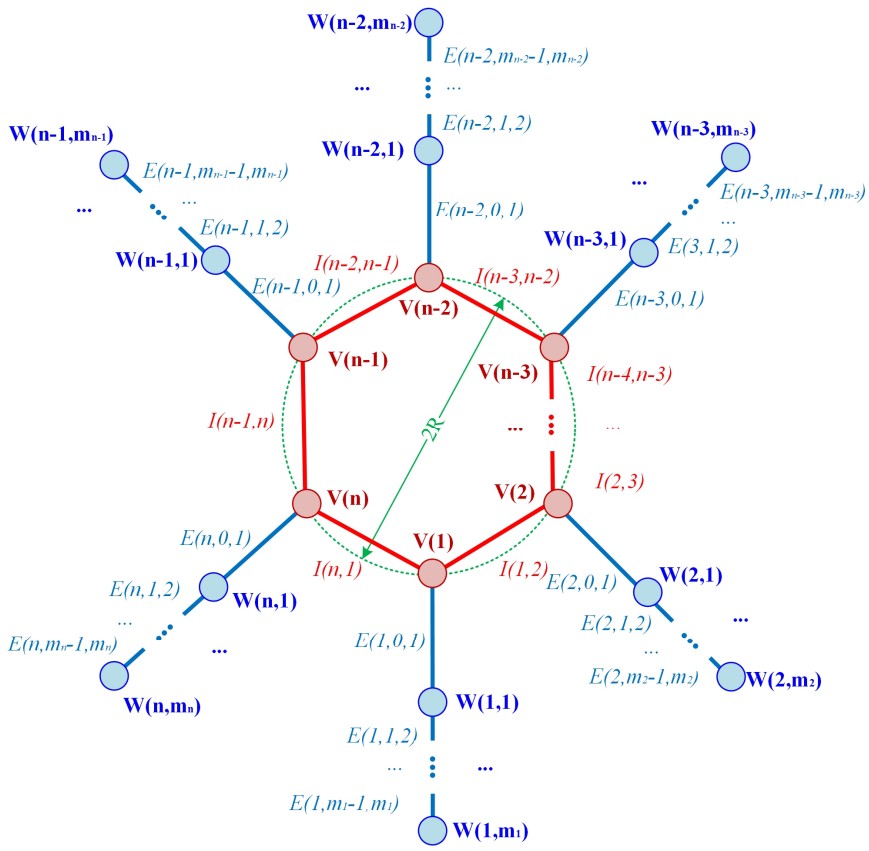

**Figure 11.** Theoretical geometrical model of the city logistics system of the concentrated sets of delivery locations.

The notation for the theoretical model can be seen below:

- $n$: number of vertices (CSDLs) on the central ring (given as polygons)
- $m_i$: number of vertices (CSDLs) on ray "$i$"
- $V(i)$: CSDL No. "$i$" on the central ring; its sum demand is $Q_{V(i)}$
- $W(i,j)$: CSDL No. "$j$" on ray "$i$"; its sum demand is $Q_{W(i,j)}$ (based on this logic, the CSDL on the central ring can be marked as $W(i,0)$
- $I(i,i+1)$: the edge (road section) between $V(i)$ and $V(i+1)$ on the central ring; its length is $D[I(i,i+1)]$, $i = 1 \ldots n$; $I(n,n+1) = I(n,1)$

- $E(i, j-1, j)$: the edge (road section) between $W(i, j-1)$ and $W(i, j)$ on ray "$i$"; its length is $D[E(i, j-1, j)]$, $j = 1 \ldots m_i$
- $R$: the ray of the best fitting circle of the polygon of the central ring

In the theoretical model, the length of ray "$i$" can be calculated by the Formula (1):

$$d_i = \sum_{j=1}^{m_i} D[E(i, j-1, j)] \tag{1}$$

If the length of all rays is known, the average length of a ray in the examined structure can be calculated by the Formula (2):

$$D = \frac{\sum_{i=1}^{n} d_i}{n} \tag{2}$$

The average number of CSDLs on one average ray can be calculated by the Formula (3):

$$\overline{m} = \frac{\sum_{i=1}^{n} m_i}{n} \tag{3}$$

The average demand of an average CSDL on the central ring can be calculated by the Formula (4):

$$\overline{Q}_V = \frac{\sum_{i=1}^{n} Q_{V(i)}}{n} \tag{4}$$

The average demand of an average CSDL on ray "$i$" can be calculated by the Formula (5):

$$\overline{Q}_{W(i)} = \frac{\sum_{j=1}^{m_i} Q_{W(i,j)}}{m_i} \tag{5}$$

The average demand of an average CSDL of the examined geometrical model can be calculated by the Formula (6):

$$\overline{Q} = \frac{\sum_{i=1}^{n} Q_{V(i)} + \sum_{i=1}^{n} \sum_{j=1}^{m_i} Q_{W(i,j)}}{n + \sum_{i=1}^{n} m_i} \tag{6}$$

### 3.3. The Symmetrical Geometrical Model

Next, the symmetrical geometrical structure is defined, which can be seen in Figure 12, with the main parameters of the full symmetrical case.

The main properties of the symmetrical case can be seen below:

- the number of the CSDLs on the rays is the same on every ray *(m)*
- the length of the $I(i-1, i)$ edges is the same for every "$i$" value ($I = 1 \ldots n$), so the central ring will be a regular polygon
- the length of the $E(i, j-1, j)$ edge is the same on every ray, for every "$i$" and "$j$" value ($i = 1 \ldots n$; $j = 1 \ldots m$)
- the demands are the same for every CSDL, as in Formula (7):

$$\overline{Q} = Q_{V(i)} = \cdots = Q_{V(n)} = Q_{W(1,1)} = \cdots = Q_{W(i,j)} = \cdots = Q_{W(n,m)} \tag{7}$$

It can be assumed because of the symmetry that this symmetrical model will give later the reference geometrical structure that can be used as a benchmark. During the future comparison, it will be essential to get an answer as to what percentage of the current examined model approximates the reference structure, and based on this, it will be possible to decide on the suitability of the examined cargo bike-based concept for the given urban area. In the current case, the reference geometrical model will be a structure that is optimal in terms of well-defined aspects (e.g., sum mileage, the necessary number of vehicles, energy consumption, emission). It is important that in this case, we are looking for a

reference model of given CSDLs and their relative position to each other, so a reference is needed that is adapted to the specifics of the examined system (e.g., in the case of a model with 6 central CSDLs and 20 CSDLs of the rays, we need a reference structure with 6 central CSDLs and 20 CSDLs on the rays). Later, in the next phases of the research, we would like to prove the reference structure in a heuristic way by use of the presented simulation model and in a mathematical way, too. It will be an essential question to find out if one "optimal" reference structure is there only or if it is possible to find alternative solutions as well.

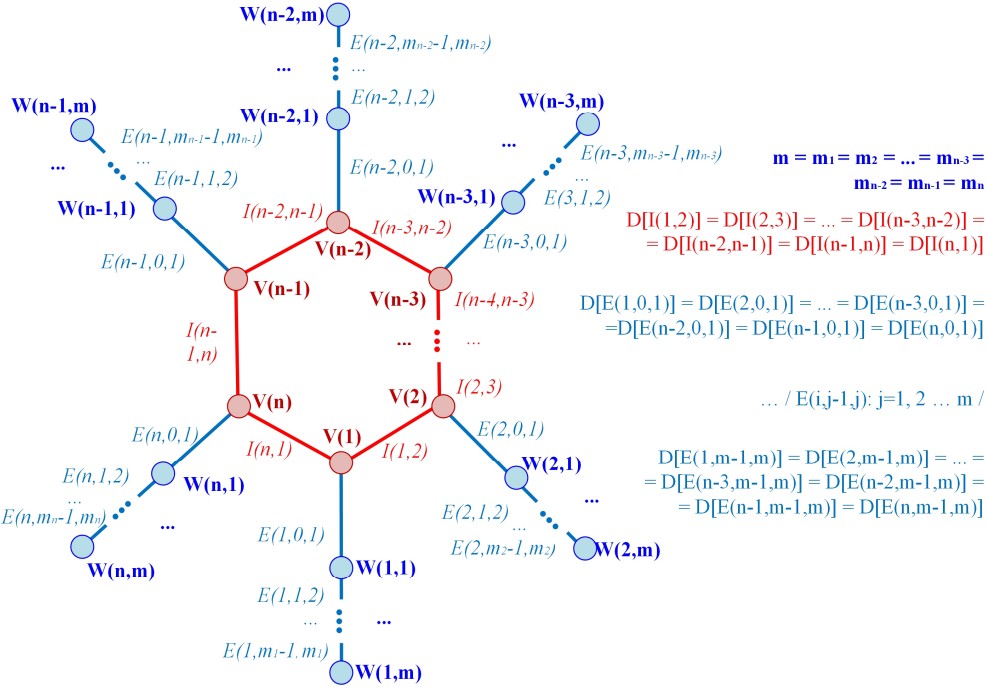

**Figure 12.** The symmetrical geometrical model of the city logistics system of the concentrated sets of delivery locations.

### *3.4. Geometrical Model of the Real Network*

The real geometrical models are significantly different from the symmetrical model in every city logistics system; so, in real systems, the statements above are not true, the model is distorted, such as in the case of the geometrical model of the shopping malls in Budapest. In these real systems, there are also branches, i.e., certain sections of different rays are overlapping each other, and it is also possible to have central CSDLs, from which more than one ray starts. As can be seen in the example of Budapest, some sections of the rays are in the real system broken lines. The left side of Figure 13 (below) shows the model of a possible real system in the case of *n* = 6 (6 rays); the right side of the figure shows the simplified version of the model. It is important to highlight that because of the real structure of the transportation network, the rays and the central ring itself are given in the real model by broken lines, but they can be modeled as straight sections and as a polygon, with the original distances, as they are the critical indicators.

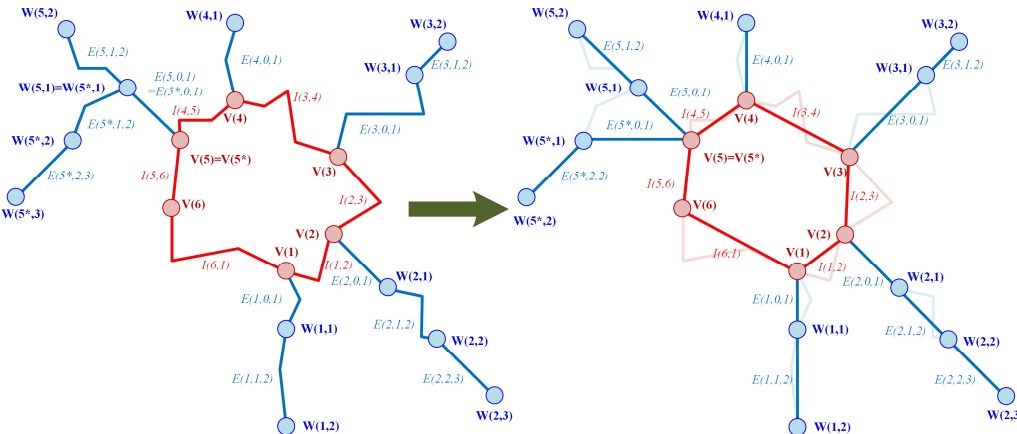

**Figure 13.** Geometrical model of a real city logistics system of the concentrated sets of delivery locations with slashes and then simplified, in the case of *n* = 6.

Thus, in these real cases, there can be such a central CSDL, from where more than one ray starts (these will be signed by the * character), and it is also possible to have CSDLs without further rays ($m_i = 0$). Thus, the number of rays will be more than, equal to, or less than the number of the central CSDLs (*n*). The general properties of such a real model can be seen below:

- the number of CSDLs on one ray is not equal in every case: in the example $m_1 = 2$, $m_2 = 3$, $m_3 = 2$, $m_4 = 1$, $m_5 = 2$, $m_{5*} = 2$, $m_6 = 0$
- the length of the $I(i-1,i)$ edges is not the same in every case
- the length of the $E(i,j-1,j)$ edges is not the same in every case; there can be differences between the rays and on the rays as well

Next, we present the geometrical model with the new notation for the preciously examined example from Budapest.

### 3.5. Application of the Geometrical Model: Results for Budapest

After developing the graph theory-based notation of the geometric model, we are going to present it in a numerical example as well. The object of the numerical example is the previously examined network from Budapest. This is a good example of the real geometrical model, as the central ring and the rays can be clearly defined, but it is distorted at the same time—there are several branches in it, and one of the central CSDLs is not exactly fitting to the central ring (SM013). It should be mentioned that in the case of three pairs of shopping malls, the distance between them is close to zero, so they could even be served together with a common cross-dock in a new system. The examined geometrical model with the graph-theory-based notation can be seen in Figure 14. To the polygon of the central ring, an ellipse can be fitted instead of a circle, its major axis is 4.7 km, and its minor one is 3.5 km. On the figure, at the branches *, **, and *** are used to show those rays which belong to the same central CSDL.

For the examination of the geometrical model of the shopping malls in Budapest, all the main indicators were calculated (distances, goods amount, number of CSDLs) and were defined and described in the previous sections. The results can be seen in Table 2.

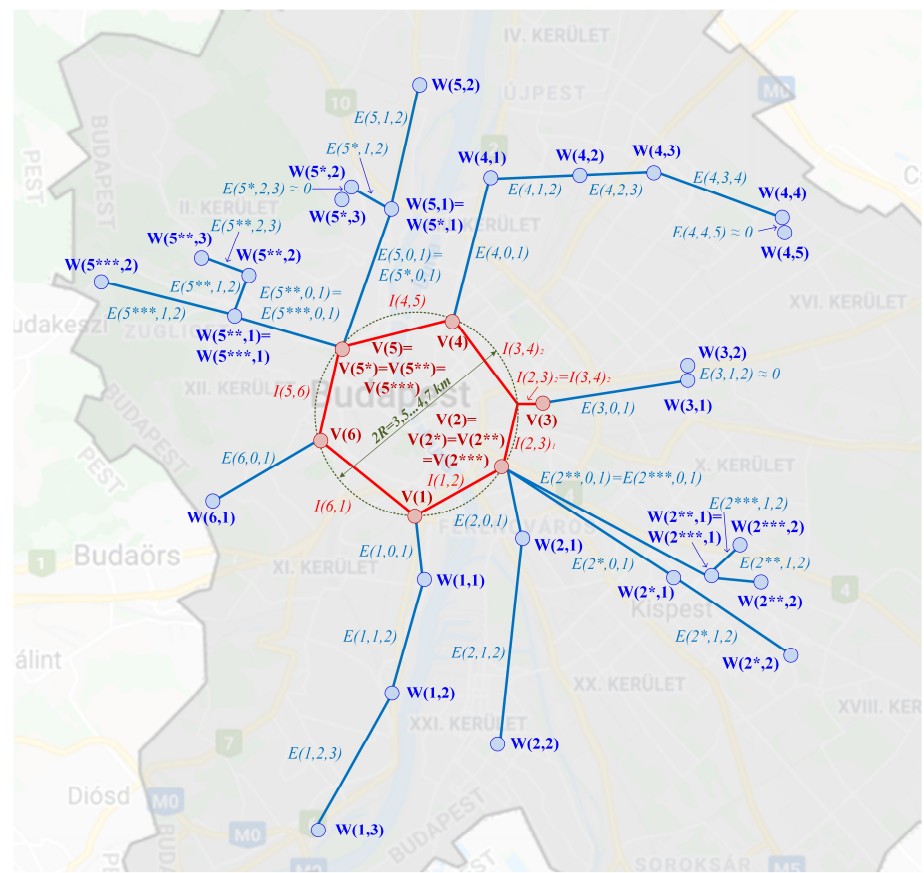

**Figure 14.** Geometrical model of the shopping malls in Budapest.

**Table 2.** Main parameters of the geometrical model of the shopping malls in Budapest.

| Ray | Length of Ray i ($d_i$) [km] | Average Length of Rays (D) [km] | Number of Concentrated Sets of Delivery Locations (CSDLs) on Ray i ($m_i$) | Average Number of CSDLs per Ray ($\bar{m}$) | Number of CSDLs on the Central Ring ($n$) | Average Demand of a CSDL on the Central Ring ($\bar{Q}_V$) [t/month] | Average Demand of a CSDL on Ray i ($\bar{Q}_{W(i)}$) [t/month] | Average Demand of a CSDL in the System ($\bar{Q}$) [t/month] |
|---|---|---|---|---|---|---|---|---|
| 1 | 8.84 | | 3 | | | | 926.4 | |
| 2 | 9.47 | | 2 | | | | 495.6 | |
| 2 * | 11.44 | | 2 | | | | 367.5 | |
| 2 ** | 8.84 | | 2 | | | | 1248.9 | |
| 2 *** | 7.54 | | 2 | | | | 1008.9 | |
| 3 | 3.80 | 7.46 | 2 | 2.42 | 6 | 1636.7 | 1600.1 | 841.3 |
| 4 | 11.10 | | 5 | | | | 822.5 | |
| 5 | 8.58 | | 2 | | | | 276.5 | |
| 5 * | 5.76 | | 3 | | | | 272.0 | |
| 5 ** | 4.62 | | 3 | | | | 330.4 | |
| 5 *** | 5.73 | | 2 | | | | 350.7 | |
| 6 | 3.85 | | 1 | | | | 296.7 | |

In Table 2, *, **, and *** are used to show those rays which belong to the same central CSDL. For example, Ray 5 is the first ray of central CSDL No. 5, 5* is its second ray, 5** is its third ray, and 5*** is its fourth ray.

The examined geometrical model is far from a symmetrical structure; additionally to the branches, the length of the rays, the number of CSDLs, and the estimated amount of goods are fluctuating. It was also examined whether any correlation could be detected between the length of the rays, the number of CSDLs on them, and the estimated average amount of goods, but the analysis showed no correlation; presumably, much more traffic characteristics, the size of the CSDLs, and the degree of concentration may be related to the quantities of goods.

Thus, we developed the theoretical, the reference, and the real geometric model of the city logistics system of the CSDLs and its notation based on graph theory, which was also applied to the examined Budapest example. Earlier, it was concluded, based on the simulation results using the specified geometrical model, that the cargo bike-based city logistics concept can be a right development direction because by integrating a relatively small number of cargo bikes into the system, many stores could be served. Additionally, a significant share of all freight demand in such a system could be served by using the bike path network instead of the already congested urban road network. By developing this complex model, and its exact notation, it became possible to examine the similar city logistics system in an exact way, and it became possible to examine that reference geometrical model in the future, which can later function as a benchmark in similar developments. Naturally, there are several further questions in this research which we would like to present next.

## 4. Discussion and Future Research

Previously, a simulation model was developed to examine a geometrical model-based new city logistics approach where cargo bikes were used in the logistics system; the theoretical model, the symmetrical model, and the model of the real system were developed, and this model was applied for the current network of the shopping malls of Budapest. Such an approach can be very useful in the future to plan new cargo bike-based city logistics systems and to integrate the use of cargo bikes into smart cities.

The primary purpose in the next steps is to define the reference geometrical model, which could be used as a benchmark in the future for the examination of any similar city logistics system. Knowing this reference geometrical structure, it will be possible to determine to what extent the reference structure approximates a given model, so it will be possible to see to what extent it is worthwhile to apply this geometric model-based city logistics approach for a given urban area. First, the plan is to define this structure by use of the presented simulation model in a heuristic way, and later we would like to prove its optimality in a mathematical way. There can be many aspects to define the reference structure; the proper selection of the objective function will be critical. It seems clear from the previous studies that multi-criteria examination will be necessary. Some ideas were already formulated (see Figure 15) for defining the reference geometrical model, along which the next research phase will be able to get started. The plan is to estimate the central ring with the best fitting circle/ellipse, so in the reference structure, the CSDLs will be moved to this circle; adding the rays to this, we are going to look after the smallest total squared deviation between the original and the new positions of the CSDLs, by rotation of the best fitting circle and the rays. In this model, the CSDLs can be moved to other rays if that would give the smallest deviation. Additionally, the optimal position of the consolidation center in the examined structure and the optimal number of consolidation centers will be important questions, too.

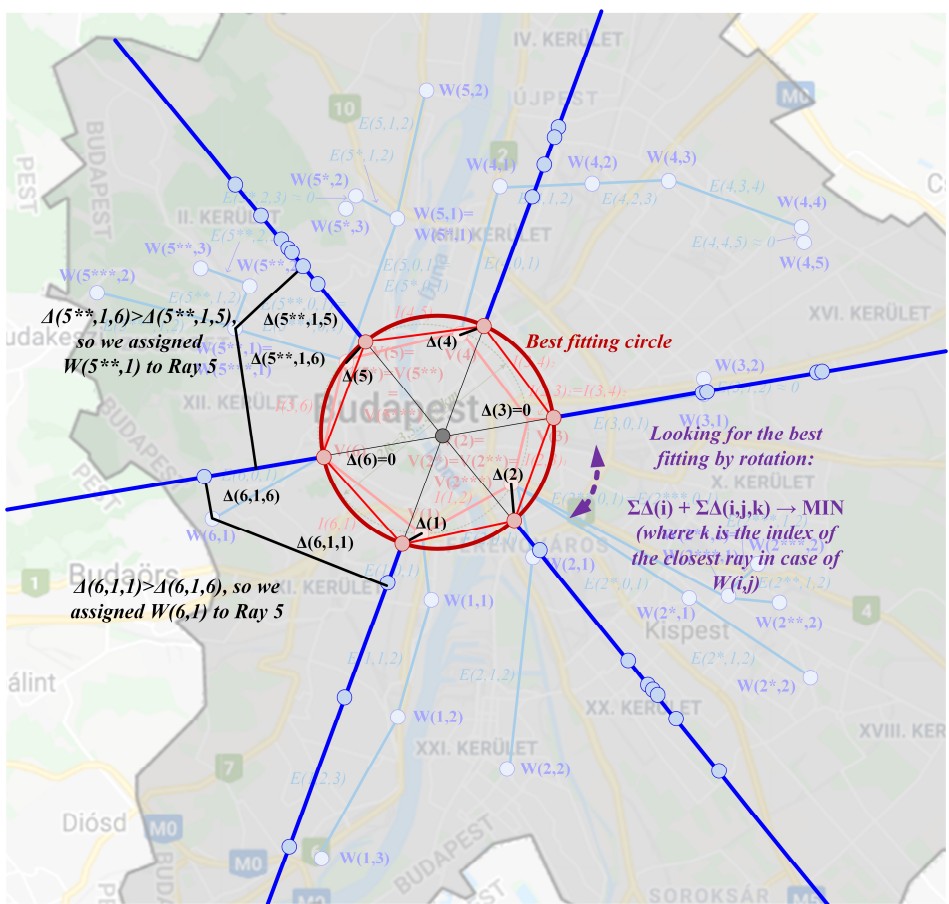

**Figure 15.** Designation of the reference geometrical model.

In this paper, primarily the case of Budapest and the theoretical model were examined, but in the future, the examination of other cities will be important too, and we would like to develop a ranking model which can help to decide, in the case of a given city, if it is worth implementing this geometrical model-based thinking or not. For this, the reference model will be able to help us. For some European cities, the examinations are already started, and based on them, Vienna and Warsaw seem to be good options for a similar concept. For this research, the analysis of the current logistics system and the estimation of the goods amounts will be very important for them, as based on our results, for example, in the case of Vienna, the characteristics of the shopping malls are much different to Budapest, and this must be considered at the estimation of delivery demands in the model.

It should be highlighted that in the case of the cities examined so far, there was no opportunity to evaluate the suitability precisely, so in the future, it will be very important to develop a ranking methodology to decide whether it is worthwhile to study such concepts in each city. Such a multicriterial method could significantly help in the future the application of this approach for a given city. Of course, in such a model, several indicators should be considered, and these indicators can have two main groups: geometrical model-based indicators and technology-based indicators.

In the case of the geometrical model-based indicators, the urban structure, the area to serve, the number of CSDLs, the properties of the model, and the network-related entropy-type indicators can be interesting. If the properties of the geometrical model will be examined, the length of the edges (the distance between the examined CSDLs on the central ring and on the rays), the covered central area, the number of CSDLs per rays, the number of rays, their length, and the number of CSDLs per rays should be highlighted. Here, it is also very important to consider the different characteristics of the different CSDLs of different cities, the different traffic demands, and their deviation as well. Later, if the

currently two-dimensional geometrical model will be developed into a three-dimensional model, the terrain conditions (slopes) of the examined urban area will be important, it will be necessary to examine it in the ranking, with considering the altitude difference between the CSDLs on the central ring and on the rays.

The other group to examine is the group of technology-based indicators. In the ranking methodology, exact, modeled cargo bike types should be examined instead of real types. In general, the most important indicators are the capacity, interchangeability of the cargo boxes, and the range (with or without assistance). In the case of a cargo bike without electric power assistance, the human factor must be considered as well, and the examination of the meteorological conditions is important too, as in the case of worse weather conditions, the open structure of the vehicles can have several disadvantages. If the terrain conditions are going to be examined as well, the capability of these modeled cargo bike types to handle slopes must be examined. For this evaluation, the delivery technologies can have three main groups:

- two- or three-wheeled cargo bikes without electric power assistance;
- two-, three-, or four-wheeled cargo bikes (with or without trailer) with electric power assistance;
- non-pedal small electric delivery vehicles (e.g., cargo mopeds; they must be considered as their capabilities and advantages are very similar to cargo bikes)

In such a ranking methodology, several indicators must be examined, so complex multicriterial models are needed in the future. Primarily, the plan is to use the Analytic Hierarchy Process method, as it is appropriate to handle such complex problems. In the next steps of the research, the exact definition of the criteria and the formalization of the ranking model will be the primary tasks. Regarding our plans, this model will be able to show the suitability of given cities for a geometrical model-based city logistics approach with the use of cargo bikes by comparing them with their reference model.

In the future, it will also be essential to examine how to integrate the railway and waterway transport modes next to the road transport into the concepts of geometrical model-based thinking. In Figure 16 below, the previously examined system in Budapest has good railway and waterway transport connections, and the urban railways are even not marked—they can give lots of extra possibilities.

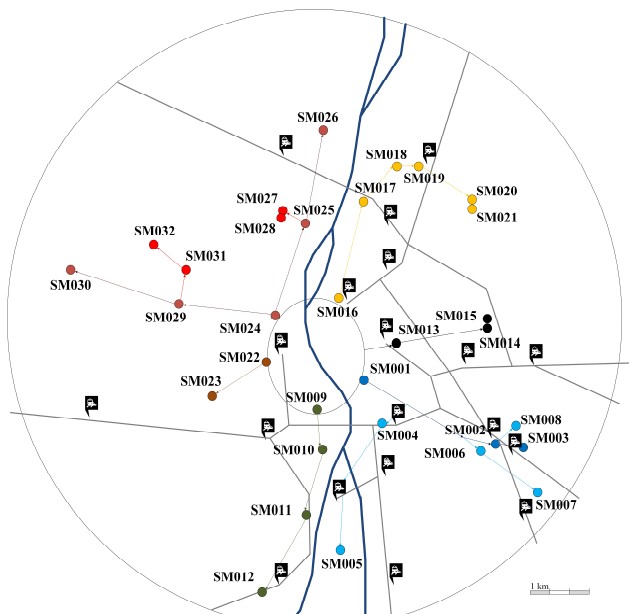

**Figure 16.** The geometrical model of Budapest, with railway and waterway connections.

In this paper, the city logistics system of the concentrated sets of delivery locations were examined, and only the business-to-business deliveries were included, from the suppliers to the stores of the CSDLs, via the consolidation center, using electric trucks and cargo bikes. Additionally, it can be very important to examine the other delivery types, primarily the business-to-customer deliveries (home deliveries, deliveries to parcel stations, or to parcel pick up points), but the deliveries between the stores of the same company and the deliveries to services should be examined as well. For these, the geometrical model and the simulation model must be developed; this can be a significant step for future research, as based on the analysis of the data from Budapest (3 shopping malls and 1 shopping area), 135 of 477 stores have business-to-customer deliveries (28%), 116 from 433 stores have incoming deliveries from other stores (27%), and 171 from 464 stores have outgoing deliveries to other stores (37%). If we would like to examine the business-to-customer deliveries, it will be important to investigate the so-called micro-hubs [51], as they can have a significant role in the geometrical model-based city logistics system with business-to-customer deliveries. Figure 17 shows those delivery transactions which must be added to the model in future research: the deliveries between the stores of the shopping malls by red lines and the home deliveries by blue lines; their geometrical model, graph-theory-based description, and the possible role of micro-hubs in the deliveries must be examined in the next steps.

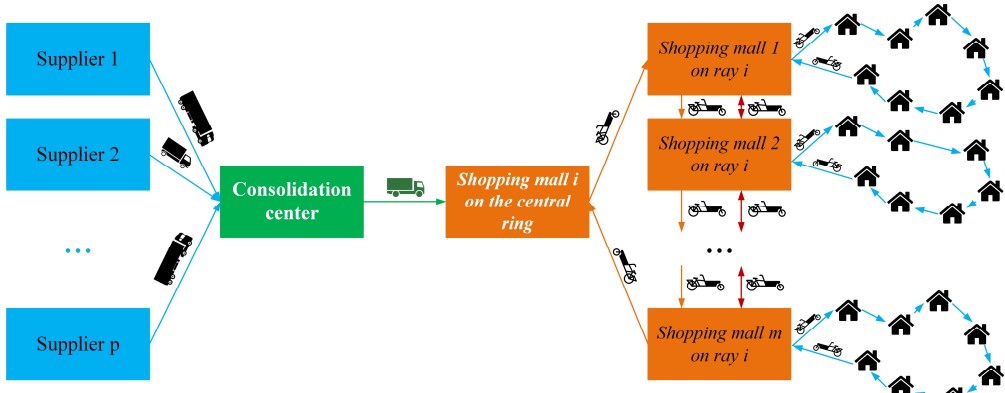

**Figure 17.** The examined concept with the deliveries between stores and home deliveries.

As a summary of the results of the paper, in Figure 18, the main steps of the application of the geometrical model-based new approach can be seen.

In this research, the development of the graph theory-based model and notation was motivated by the results provided by a previously developed simulation model. In future projects, the first steps will be to describe the geometrical model of the examined real system and to collect data about the CSDLs; based on these, it will be possible to develop a simulation model, and the results of the simulation together with the geometrical model of the real system and the reference geometrical model can provide data for the AHP-based multicriterial ranking which makes it possible to decide about the application of the new approach for given cities.

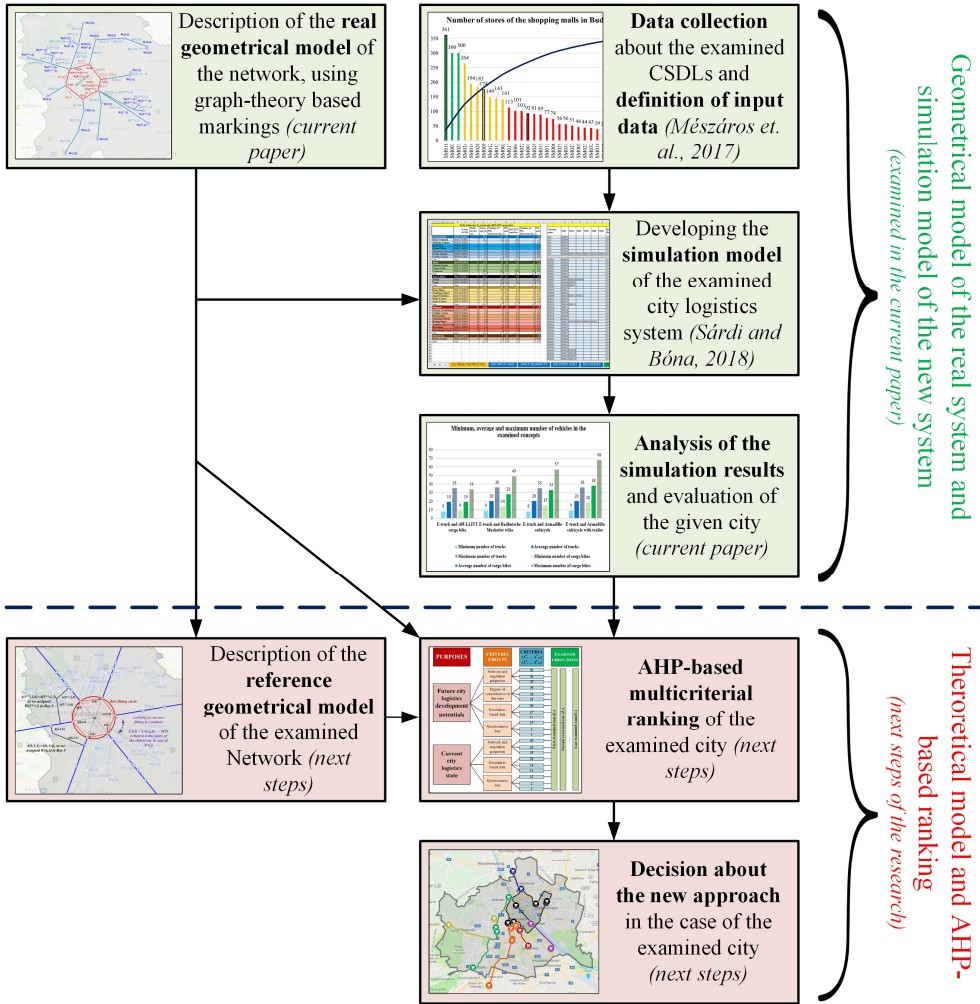

**Figure 18.** Application of the geometrical model-based approach.

## 5. Conclusions

In this paper, we examined geometrical model-based thinking in organizing city logistics by use of cargo bikes, in the logistics system of shopping malls, primarily for Budapest. This solution seems to be a new approach, based on the results of the literature review. Nowadays, these cargo bikes are more and more popular, especially in Europe, but in America and in Asia, they are playing an increasingly more significant role in city logistics. Such research is very important not only because of the current city logistics challenges but because of their possible integration to smart cities as well.

In the City Logistics Research Group of our Department [1], we are focusing on the so-called concentrated sets of delivery locations in our research project, and in this paper, we examined how cargo bikes can be used in their city logistics system based on the geometrical model of the logistics network. First, we presented the existing cargo bike-based systems and the related research results; next, our previous project was presented (from 2017). The results of this project gave the idea of this paper. In the concept, which was defined in this project, the CSDLs are divided into two groups: CSDLs on a central ring and CSDLs on rays, which start from the central ring, and we examined how to integrate the deliveries of the ones on the rays to the deliveries of the central ones.

For this research, the previously used simulation model was developed further and extended to the whole of Budapest (with all its shopping malls) by examining several different concepts using cargo bikes. Based on the results, it was considered that with a relatively small extra investment (so, with a relatively small number of additional cargo bikes), a significant amount of goods and a significant number of stores could be integrated into the new system, while the delivery costs will increase only 20–40%, and we can exclude the potential negative effects of the traffic jams on the rays (as they can cause longer delivery times, with more delivery costs). It is also important that it is simpler to hire bike couriers nowadays, and there are not enough available, suitably qualified drivers for trucks and lorries, so it can be possible that the logistics providers will be forced to use more cargo bikes; in this case, this geometrical model-based approach can help to organize their deliveries.

Next, the theoretical geometrical model, the symmetrical model, and the model of the real system were developed; with their graph theory-based notation, the description and application of such geometrical model-based solutions is an entirely new approach in city logistics planning. We applied this graph theory-based solution for the previously examined geometrical model in Budapest as a numerical example. In general, based on this geometrical model, it will be possible in the future to examine the suitability of given cities for implementation of the examined geometrical model-based new approach with cargo bikes.

In the last section of the paper, the planned next steps of the research project were presented. The first and most important task is to define the reference structure. We gave the related plans based on the example of Budapest. It is assumed that the symmetrical model can be the reference one, but alternative optimal solutions must be examined as well. Next, we would like to develop a ranking model, which will be able to rank given cities after comparing them with their reference models. This ranking model will be able to decide about the suitability of the examined cargo bike-based city logistics concepts for given cities.

The main result of this research project is the theoretical, symmetrical, and real geometrical model of the city logistics system of the concentrated sets of delivery locations, which provides an entirely new approach to the development of cargo bikes-based city logistics systems and gives the basis of our next research steps.

**Author Contributions:** Conceptualization, D.L.S. and K.B.; Data curation, D.L.S.; Formal analysis, D.L.S.; Investigation, D.L.S.; Methodology, D.L.S. and K.B.; Resources, D.L.S.; Software, D.L.S.; Supervision, K.B.; Validation, K.B.; Visualization, D.L.S.; Writing—original draft, D.L.S. and K.B. All authors have read and agreed to the published version of the manuscript.

**Funding:** This research received no external funding.

**Institutional Review Board Statement:** Not applicable.

**Informed Consent Statement:** Not applicable.

**Data Availability Statement:** Data is contained within the article and its appendices.

**Conflicts of Interest:** The authors declare no conflict of interest.

## Appendix A. Specific Values for the Three Examined Categories

**Table A1.** Estimated specific values for the three examined categories.

| Estimated Specific Parameters of the Categories | | A | B | C |
|---|---|---|---|---|
| **Number of deliveries/store** | Daily | 0.52 | 0.79 | 0.81 |
| | Weekly | 3.67 | 5.52 | 5.68 |
| | Monthly | 15.23 | 23.14 | 23.95 |
| | Yearly | 192.76 | 288.85 | 296.53 |
| **The daily amount of goods/store** | Daily minimum [kg] | 28.57 | 163.83 | 102.29 |
| | Daily maximum [kg] | 129.35 | 843.17 | 404.95 |
| | Daily minimum [m$^3$] | 1.20 | 3.59 | 2.00 |
| | Daily maximum [m$^3$] | 3.66 | 10.43 | 6.29 |
| **The monthly amount of goods/store** | Monthly minimum [kg] | 814.71 | 4851.19 | 3027.25 |
| | Monthly maximum [kg] | 3689.10 | 24,932.04 | 11,958.33 |
| | Monthly minimum [m$^3$] | 34.47 | 105.24 | 58.48 |
| | Monthly maximum [m$^3$] | 105.58 | 306.29 | 183.65 |
| **Necessary number of cargo bike deliveries/store** | eBULLITT/day | 0.11 | 0.11 | 0.04 |
| | eBULLITT/month | 3.22 | 3.02 | 1.03 |
| | Trike/day | 0.23 | 0.20 | 0.14 |
| | Trike/month | 6.43 | 5.72 | 3.90 |
| | Armadillo/day | 0.19 | 0.17 | 0.11 |
| | Armadillo/month | 5.42 | 4.96 | 3.04 |
| | Armadillo with trailer/day | 0.23 | 0.21 | 0.10 |
| | Armadillo with trailer/month | 6.51 | 6.06 | 2.73 |
| **Expected share of stores, they can be served by cargo bikes/store** | eBULLITT | 25.8% | 20.3% | 7.5% |
| | Trike | 63.4% | 55.4% | 32.5% |
| | Armadillo | 52.7% | 45.9% | 25.0% |
| | Armadillo with trailer | 71.0% | 63.5% | 32.5% |
| **Amount of goods to be delivered by cargo bikes/store** | eBULLITT daily [kg] | 1.09 | 0.26 | 0.05 |
| | eBULLITT monthly [kg] | 30.73 | 7.32 | 1.53 |
| | Trike daily [kg] | 11.23 | 5.83 | 4.44 |
| | Trike monthly [kg] | 321.10 | 167.30 | 124.88 |
| | Armadillo daily [kg] | 6.32 | 4.73 | 3.74 |
| | Armadillo monthly [kg] | 178.16 | 135.79 | 105.38 |
| | Armadillo with trailer daily [kg] | 13.70 | 8.86 | 4.44 |
| | Armadillo with trailer monthly [kg] | 391.69 | 253.41 | 124.88 |

# Appendix B. Number of Stores, Deliveries and Amounts of Goods

**Table A2.** Number of stores, deliveries, and amounts of goods.

| ID | Category | Number of Stores | Number of Deliveries | | | |
|---|---|---|---|---|---|---|
| | | | Daily | Weekly | Monthly | Yearly |
| **SM001** | C | 92 | 68 | 470 | 1984 | 24,553 |
| **SM002** | B | 141 | 100 | 701 | 2936 | 36,656 |
| **SM003** | A | 300 | 142 | 992 | 4111 | 52,047 |
| **SM004** | C | 101 | 74 | 516 | 2178 | 26,955 |
| **SM005** | C | 46 | 34 | 235 | 992 | 12,277 |
| **SM006** | C | 74 | 54 | 378 | 1596 | 19,749 |
| **SM007** | C | 35 | 26 | 179 | 755 | 9341 |
| **SM008** | C | 19 | 14 | 98 | 410 | 5071 |
| **SM009** | B | 174 | 124 | 865 | 3624 | 45,235 |
| **SM010** | C | 39 | 29 | 200 | 841 | 10,409 |
| **SM011** | C | 77 | 57 | 394 | 1660 | 20,550 |
| **SM012** | B | 149 | 106 | 741 | 3103 | 38,735 |
| **SM013** | B | 143 | 102 | 711 | 2978 | 37,176 |
| **SM014** | B | 194 | 138 | 964 | 4040 | 50,434 |
| **SM015** | C | 89 | 65 | 455 | 1919 | 23,752 |
| **SM016** | A | 361 | 171 | 1194 | 4947 | 62,629 |
| **SM017** | C | 113 | 83 | 578 | 2436 | 30,157 |
| **SM018** | C | 23 | 17 | 118 | 496 | 6139 |
| **SM019** | C | 20 | 15 | 103 | 432 | 5338 |
| **SM020** | B | 183 | 130 | 910 | 3811 | 47,574 |
| **SM021** | A | 300 | 142 | 992 | 4111 | 52,047 |
| **SM022** | C | 101 | 74 | 516 | 2178 | 26,955 |
| **SM023** | C | 44 | 33 | 225 | 949 | 11,743 |
| **SM024** | B | 264 | 188 | 1312 | 5498 | 68,632 |
| **SM025** | C | 56 | 41 | 287 | 1208 | 14,945 |
| **SM026** | C | 26 | 19 | 133 | 561 | 6939 |
| **SM027** | C | 51 | 38 | 261 | 1100 | 13,611 |
| **SM028** | C | 14 | 11 | 72 | 302 | 3737 |
| **SM029** | C | 91 | 67 | 465 | 1962 | 24,286 |
| **SM030** | C | 13 | 10 | 67 | 281 | 3470 |
| **SM031** | C | 56 | 41 | 287 | 1208 | 14,945 |
| **SM032** | C | 43 | 32 | 220 | 927 | 11,476 |

Table A3. Estimated daily and the monthly amount of goods for the examined shopping malls.

| ID | The Daily Amount of Goods | | | | The Monthly Amount of Goods | | | |
|---|---|---|---|---|---|---|---|---|
| | Daily Min. [kg] | Daily Max. [kg] | Daily Min. [m$^3$] | Daily Max. [m$^3$] | Monthly Min. [kg] | Monthly Max. [kg] | Monthly Min. [m$^3$] | Monthly Max. [m$^3$] |
| SM001 | 8470 | 33,530 | 166 | 521 | 250,656 | 990,150 | 4843 | 15,207 |
| SM002 | 20,791 | 106,999 | 456 | 1324 | 615,617 | 3,163,876 | 13,356 | 38,869 |
| SM003 | 7715 | 34,926 | 324 | 990 | 219,971 | 996,057 | 9307 | 28,507 |
| SM004 | 9298 | 36,810 | 182 | 572 | 275,177 | 1,087,012 | 5317 | 16,694 |
| SM005 | 4235 | 16,765 | 83 | 261 | 125,328 | 495,075 | 2422 | 7604 |
| SM006 | 6813 | 26,970 | 134 | 419 | 201,615 | 796,425 | 3895 | 12,231 |
| SM007 | 3222 | 12,756 | 63 | 199 | 95,359 | 376,688 | 1843 | 5785 |
| SM008 | 1750 | 6925 | 35 | 108 | 51,766 | 204,488 | 1001 | 3141 |
| SM009 | 25,656 | 132,042 | 563 | 1633 | 759,697 | 3,904,358 | 16,482 | 47,965 |
| SM010 | 3591 | 14,214 | 71 | 221 | 106,257 | 419,738 | 2053 | 6447 |
| SM011 | 7089 | 28,063 | 139 | 436 | 209,789 | 828,712 | 4053 | 12,727 |
| SM012 | 21,970 | 113,070 | 482 | 1399 | 650,545 | 3,343,387 | 14,114 | 41,074 |
| SM013 | 21,086 | 108,517 | 463 | 1343 | 624,349 | 3,208,754 | 13,545 | 39,420 |
| SM014 | 28,605 | 147,219 | 627 | 1821 | 847,019 | 4,353,134 | 18,376 | 53,479 |
| SM015 | 8194 | 32,437 | 161 | 504 | 242,483 | 957,862 | 4685 | 14,711 |
| SM016 | 9283 | 42,027 | 389 | 1191 | 264,698 | 1,198,588 | 11,199 | 34,303 |
| SM017 | 10,403 | 41,184 | 204 | 640 | 307,871 | 1,216,162 | 5948 | 18,678 |
| SM018 | 2118 | 8383 | 42 | 131 | 62,664 | 247,538 | 1211 | 3802 |
| SM019 | 1842 | 7290 | 36 | 114 | 54,491 | 215,250 | 1053 | 3306 |
| SM020 | 26,984 | 138,871 | 592 | 1718 | 798,992 | 4,106,307 | 17,334 | 50,446 |
| SM021 | 7715 | 34,926 | 324 | 990 | 219,971 | 996,057 | 9307 | 28,507 |
| SM022 | 9298 | 36,810 | 182 | 572 | 275,177 | 1,087,012 | 5317 | 16,694 |
| SM023 | 4051 | 16,036 | 80 | 250 | 119,879 | 473,550 | 2316 | 7273 |
| SM024 | 38,927 | 200,339 | 853 | 2478 | 1,152,644 | 5,923,853 | 25,007 | 72,775 |
| SM025 | 5156 | 20,410 | 101 | 318 | 152,574 | 602,700 | 2948 | 9256 |
| SM026 | 2394 | 9476 | 47 | 148 | 70,838 | 279,825 | 1369 | 4298 |
| SM027 | 4695 | 18,588 | 92 | 289 | 138,951 | 548,888 | 2685 | 8430 |
| SM028 | 1289 | 5103 | 26 | 80 | 38,144 | 150,675 | 737 | 2314 |
| SM029 | 8378 | 33,166 | 164 | 516 | 247,932 | 979,387 | 4790 | 15,041 |
| SM030 | 1197 | 4738 | 24 | 74 | 35,419 | 139,913 | 685 | 2149 |
| SM031 | 5156 | 20,410 | 101 | 318 | 152,574 | 602,700 | 2948 | 9256 |
| SM032 | 3959 | 15,672 | 78 | 244 | 117,155 | 462,788 | 2264 | 7108 |

**Table A4.** Suitability of cargo bikes, the sum, and the average amounts.

| | | Sum | Average (for One Shopping Mall) | Average (for One Store) |
|---|---|---|---|---|
| **Number of stores** | | 3432 | 107.3 | 1 |
| **Cargo bike deliveries** | eBULLITT/day | 258.3 | 8.1 | 0.1 |
| | eBULLITT/month | 7310.5 | 228.5 | 2.1 |
| | Trike/day | 570.2 | 17.8 | 0.2 |
| | Trike/month | 16,269.7 | 508.4 | 4.7 |
| | Armadillo/day | 477.3 | 14.9 | 0.1 |
| | Armadillo/month | 13,605.4 | 425.2 | 4.0 |
| | Armadillo with trailer/day | 540.3 | 16.9 | 0.2 |
| | Armadillo with trailer/month | 15,451.9 | 482.9 | 4.5 |
| **The average number of stores to be served by cargo bikes** | eBULLITT | 548 | 17.1 | 16.0% |
| | Trike | 1545 | 48.3 | 45.0% |
| | Armadillo | 1263 | 39.5 | 36.8% |
| | Armadillo with trailer | 1699 | 53.1 | 49.5% |
| **Amount of goods to be delivered by cargo bikes** | eBULLITT daily [kg] | 1290.1 | 40.3 | 0.4 |
| | eBULLITT daily [m$^3$] | 63.3 | 2.0 | 0.0 |
| | eBULLITT monthly [kg] | 36,477.1 | 1139.9 | 10.6 |
| | eBULLITT monthly [m$^3$] | 1790.9 | 56.0 | 0.5 |
| | Trike daily [kg] | 21,146.0 | 660.8 | 6.2 |
| | Trike daily [m$^3$] | 743.6 | 23.2 | 0.2 |
| | Trike monthly [kg] | 603,087.1 | 18,846.5 | 175.7 |
| | Trike monthly [m$^3$] | 21,214.5 | 663.0 | 6.2 |
| | Armadillo daily [kg] | 14,900.1 | 465.6 | 4.3 |
| | Armadillo daily [m$^3$] | 427.4 | 13.4 | 0.1 |
| | Armadillo monthly [kg] | 422,597.9 | 13,206.2 | 123.1 |
| | Armadillo monthly [m$^3$] | 12,185.2 | 380.8 | 3.6 |
| | Armadillo with trailer daily [kg] | 26,689.0 | 834.0 | 7.8 |
| | Armadillo with trailer daily [m$^3$] | 998.5 | 31.2 | 0.3 |
| | Armadillo with trailer monthly [kg] | 760,858.2 | 23,776.8 | 221.7 |
| | Armadillo with trailer monthly [m$^3$] | 28,549.1 | 892.2 | 8.3 |

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
