# Peer review of "A Geometrical Structure-Based New Approach for City Logistics System Planning with Cargo Bikes and Its Application for the Shopping Malls of Budapest"

_applsci, doi:10.3390/app11083300_

Round 1
Reviewer 1 Report
I consider that this paper presents an interesting approach, with results that could be useful for the public and private sector... from the detailed reading of the article, the following comments arouse:
Do the authors only focus on their research? What about the other researches?
I see only one reference from this journal, so, why the authors consider that publish in applied science is the best option?
I recommend that authors rewrite the abstract, I consider it irrelevant to say: “Some years ago, we developed a concept where a significant part of the deliveries of shopping malls was organized with electric trucks and cargo bikes. This previous project led to the idea of this paper to examine the application of cargo bikes in the logistics system of the urban concentrated sets of delivery locations in general with an investigation of the geometrical structure of the logistics network.” (lines 15 to 19)
I believe that the text could have a more scientific tone, please try to change this.
Reviewer 2 Report
Nowadays, cargo bikes are presented as a sustainable solution for last mile delivery in cities. This solution tends to minimize GHG emissions, reduce traffic congestion and improve timeliness of deliveries. Therefore, I find the paper very interesting and worth from the theoretical and practical perspective. The paper has its application value that is presented by authors – “model that can be used in the future for city logistics system planning with cargo bikes”.
The undoubtable asset of the paper is the continuation of research presented by authors in previous articles. Such wide and continuous research in the field of sustainable solutions in last mile delivery is essential for a sustainable transition of logistics operations within cities. This is one of the main trends in the city logistics field.
Structure of the paper needs to be systemized. There is no explanation of the consecutive sections presented in the manuscript in the introduction section. The structure of the paper is widely explained in the conclusions section. Besides, there is no research question in the beginning of the paper. It appears firstly in the lines 503-505.
Concentrated sets of delivery locations refer to the micro-hubs as well if we take the B2C point of view. I am fully aware that authors are focusing in their research on B2B deliveries to single delivery locations or concentrated sets of delivery locations, however, it should be worthful to verify the model in B2C deliveries in CEP-sector (home deliveries, parcel locker, pick-up/drop-off points, etc.). Especially from the perspective of cargo bikes delivery in the last mile, I encourage authors of the paper to examine in their future research the role of micro-hubs in city logistics system planning. Even in the current research authors stated that 20% of respondents that are in favor of cargo bikes suggest “bikes could be used for home deliveries” (section 2.3). I suggest one article and one report for consideration to future research:
- Ballare, S. and Lin, J. (2020) ‘Investigating the use of microhubs and crowdshipping for last mile delivery’, Transportation Research Procedia, 46, pp. 277–284.
- Deloison, T. et al. (2020) The Future of the Last-Mile Ecosystem. Transition Roadmaps for Public- and Private-Sector Players.
There is one thing that I would like to be briefly explained by authors: how big research sample those 5 SCDLs with 540 stores constitutes in Budapest? It is not clear if this 5 SCDLs are all the shopping centers (areas) in Budapest. The answer that is presented in line 355 is not corresponding with the information in line 56. I suggest to include that information in the introduction section.
I find very interesting from the theoretical point of view the categorization of shopping malls through ABC analysis with accordance to the potential use of cargo bikes. It is a valuable input into sustainable city logistic theory.
I couldn’t agree more with the statement: “If we consider only the financial reasons, of course, it is better to use the direct delivery solution, but if the potential effects of the traffic jams (late deliveries, extra deliveries needed with additional vehicles, etc.) are considered, the currently not colossal difference can be even smaller.” This is a good argument for city logistics stakeholders in favor for cargo bikes.
Line 388 – “shopping malls with a law number of stores” – there should be word “low”
Line 398, 401 – upper case in the word “m3” à m3
There is only one reference to the Applied Sciences Journal. What is the input of the authors to the journal and how the paper relates to the previous articles?
Concluding, the paper is very interesting and useful for academia and practice. I am extremely interested in the further analysis as authors of the paper stated in the discussion section. However, the reviewed paper needs to be systemized and revised in the following fields:
- Structure of the paper as it was stated above.
- Research sample explanation – number of analyzed SCDLs in entire population.
- References to other papers in Applied Sciences
- Small corrections in language as it is stated above (lines 388, 398, 401).
Round 2
Reviewer 2 Report
I am satisfied with the improvements made by authors in te revised manuscript.